# P2Seg: Pointly-supervised Segmentation via Mutual Distillation

**Zipeng Wang**[1][*], **Xuehui Yu**[1][*], **Xumeng Han**[1], **Wenwen Yu**[1], **Zhixun Huang**[2]
**Jianbin Jiao**[1], **Zhenjun Han**[1][†]
[1]University of Chinese Academy of Sciences  [2]Xiaomi AI Lab, Beijing, China
wangzipeng22@mails.ucas.ac.cn

## ABSTRACT

Point-level Supervised Instance Segmentation (PSIS) aims to enhance the applicability and scalability of instance segmentation by utilizing low-cost yet instance-informative annotations. Existing PSIS methods usually rely on positional information to distinguish objects, but predicting precise boundaries remains challenging due to the lack of contour annotations. Nevertheless, weakly supervised semantic segmentation methods are proficient in utilizing intra-class feature consistency to capture the boundary contours of the same semantic regions. In this paper, we design a **M**utual **D**istillation **M**odule (MDM) to leverage the complementary strengths of both instance position and semantic information and achieve accurate instance-level object perception. The MDM consists of **S**emantic to **I**nstance (S2I) and **I**nstance to **S**emantic (I2S). S2I is guided by the precise boundaries of semantic regions to learn the association between annotated points and instance contours. I2S leverages discriminative relationships between instances to facilitate the differentiation of various objects within the semantic map. Extensive experiments substantiate the efficacy of MDM in fostering the synergy between instance and semantic information, consequently improving the quality of instance-level object representations. Our method achieves 55.7 $mAP_{50}$ and 17.6 mAP on the PASCAL VOC and MS COCO datasets, significantly outperforming recent PSIS methods and several box-supervised instance segmentation competitors. Our code is available on https://github.com/ucas-vg/P2Seg-Public.

## 1 INTRODUCTION

Instance segmentation is a critical task in computer vision, where semantic segmentation estimation and instance discrimination are equally important. In recent years, instance segmentation has gained extensive interest due to its significance in various application scenarios and research domains (Dai et al., 2016a; 2015; 2016b; He et al., 2017b; Kirillov et al., 2017; Liu et al., 2016; Novotny et al., 2018) and it aims not only to locate objects accurately but also to estimate their boundaries to differentiate instances of the same category. The advanced instance segmentation methods use point annotations that only approximate object positions, making it difficult to capture detailed features and accurate boundaries. Hence, relying solely on point annotations has limitations that would not allow instance segmentation methods to provide precise object contours. On the other hand, semantic segmentation in computer vision involves image classification, object recognition, and semantic region boundary delineation, and its goal is to divide the image into distinct regions and assign category labels. Although advanced semantic segmentation excels at precise semantic region boundaries, it often struggles with instance-level discrimination within the same category.

Semantic information struggles to define precise boundaries when multiple same-class objects are close or overlapping. For example, if an image contains multiple bottles, semantic information methods lumps them into one "bottle" region, unable to distinguish between different bottles. Therefore, considering the limitations both instance information and semantic information have, it can be concluded that instance information is vital for accurate discrimination among instances but is limited

---

[*]Equal contribution.
[†]Corresponding author.

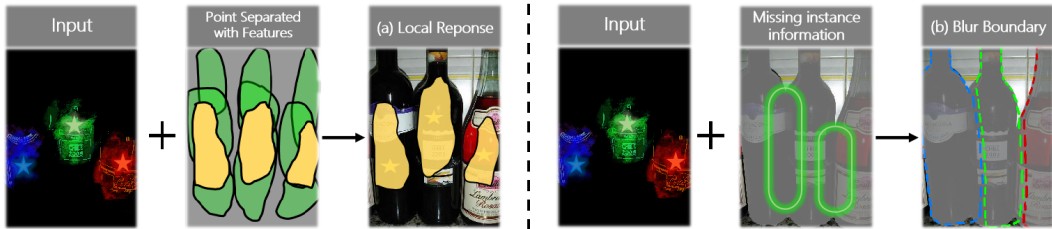

Figure 1: Two limitations of PSIS methods: **Left:** severe local responses caused by points separated with image features. **Right:** ambiguous boundaries caused by missing instance information.

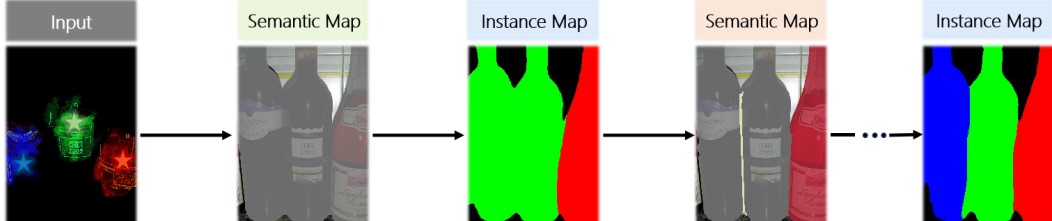

Figure 2: The illustration of S2I and I2S Complementary Advantage. Semantic and instance information is mutually distilled, resulting in high-quality instance segmentation maps.

to only instance point annotations; semantic information excels in scene understanding and object recognition but struggles with instance-level information. Both have their own merits. In summary, as shown in Fig. 1 the two problems faced by Point-level Supervised Instance Segmentation (PSIS) methods are:

- *The local response caused by the separation of points from image features.* Image features and point supervision are decoupled and heavily rely on off-the-shelf proposals for instance segmentation, resulting in localized outcomes.
- *The semantic segmentation estimation and instance differentiation are separated.* Point supervision is underutilized in the interaction of semantic segmentation and instance information, resulting in inaccurate instance estimation and challenges in distinguishing similar object boundaries.

In this paper, we aim to solve the above two problems through the Mutual Distillation Module (**MDM**) of the P2seg method, which leverages the advantages of instance and semantic segmentation. We achieve this by employing the **S**emantic to **I**nstance (S2I) and **I**nstance to **S**emantic (I2S) modules within MDM for knowledge distillation. Specifically, our contributions are as follows:

- We design an MDM module that mutually distills instance and semantic information to facilitate improved PSIS without relying on pre-trained proposals.
- We develop the S2I method to transfer semantic information to instances, guiding the network toward producing improved final instance segmentation labels. These enhanced labels lead to superior performance in the subsequent segmentor training.
- We propose the I2S module that merges instance details with class map (semantic score predicted by HRNet (Sun et al., 2019)) via affinity matrix. This integration embeds instance information into semantic context to facilitate mutual distillation and improve instance estimation accuracy.

## 2 RELATED WORK

**Weakly-Supervised Semantic Segmentation(WSSS).** Most WSSS methods extract object location using CAM from CNNs with image-level labels. However, CAM lacks precise boundaries and is insufficient for training semantic networks. ReCAM (Chen et al., 2022) refines CAM with SCE loss. CLIMS (Xie et al., 2022) uses a text-driven approach for better object activation. AFA (Ru et al., 2022) adds an attention affinity module in the transformer to refine CAM. MCTformer (Xu et al., 2022) improves CAM using patch-level pairwise affinity and transformer. L2G (Jiang et al., 2022) learns comprehensive attention knowledge from multiple maps. (Kweon et al., 2023) improves the quality of CAM by reducing the inferability between segments. WSSS focuses on pixel-level classification and cannot distinguish instances within the same category.

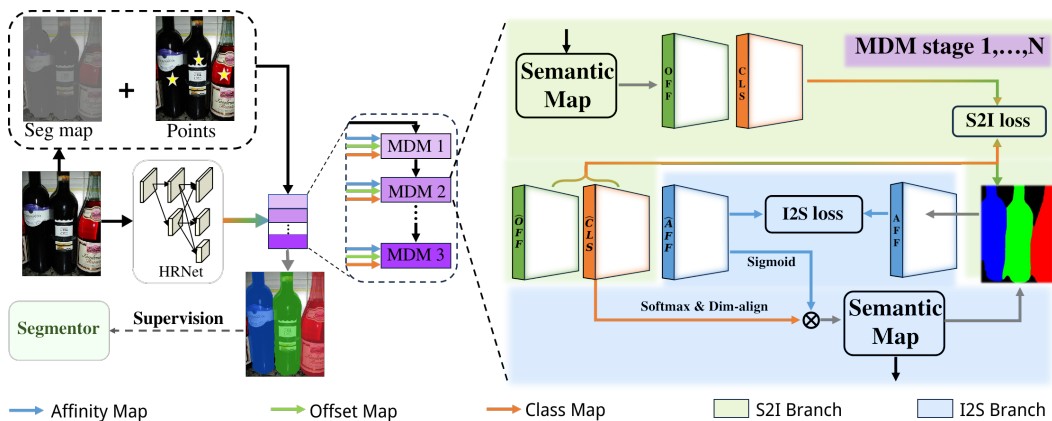

Figure 3: The architecture of MDM. In S2I branch, instance segmentation map is generated from the results of semantic segmentation using the offset map. In I2S branch, semantic segmentation results are influenced by instance segmentation map using affinity matrix. Finally, instance segmentation map generated by the trained S2I branch of MDM is used as supervision for training the segmentor.

**Weakly-Supervised Instance Segmentation(WSIS).** Weakly-supervised instance segmentation (WSIS) struggles with obtaining instance-level information from image-level labels. IRNet (Ahn et al., 2019) learns class-agnostic instance maps and pixel pair similarities, deriving pseudo instance segmentation labels without predefined proposals or extra supervision. In contrast, Label-PEnet (Ahn et al., 2019; Ge et al., 2019) transforms image-level labels into pixel-wise labels through a cascaded pipeline where the modules in this pipeline share a backbone network and use curriculum learning to enhance accuracy gradually. PDSL (Shen et al., 2021) unifies object detection and segmentation by integrating bottom-up cues into a top-down pipeline. The detection-segmentation separation allows both to perform better in their respective tasks.

**Pointly-Supervised Instance Segmentation(PSIS).** The aim of PSIS is to leverage point annotations for improved segmentation, and PSIS methods, such as WISE-Net (Laradji et al., 2020), use point-level labels. To generate high quality mask proposals, WISE-Net trains L-Net with point annotations to locate objects and then uses E-Net to group pixels, generating high quality mask proposals. BESTIE (Kim et al., 2022) refines mask proposals using point supervision and instance-level cues for robust instance segmentation. It addresses semantic drift by using self-refinement and instance-aware guidance and enhances segmentation accuracy by effectively integrating semantic and instance information, thus surpassing methods relying solely on image-level labels. SAM (Kirillov et al., 2023b) is an interactive image segmentation framework that can segment specified objects in an image using coarse hints, such as points, bounding boxes, and textual descriptions. Point2Mask (Li et al., 2023) is employed in panoramic segmentation tasks. It uses ground truth points as supervisory information to generate pseudo-label masks, thereby reducing annotation costs.

**Mutual Distillation.** Knowledge distillation (Buciluă et al., 2006; Hinton et al., 2015) distills knowledge from a complex teacher model into a simpler, compressed student model to achieve satisfactory performance. Mutual learning strategy (Zhang et al., 2018) involves collaboration among students, allowing them to learn from each other. (Wu et al., 2019) applies mutual learning to each block of a salient object detection backbone. (Zhang et al., 2019) suggests self-distillation within network. (Crowley et al., 2018) transforms a teacher network into a student network. (Hou et al., 2019) uses self-attention distillation for lane detection. (Yun et al., 2020) distills prediction distribution among same-labeled samples. (Wei et al., 2020) employs mutual learning for image classification with noisy labels, aiming to reduce network diversity during training. (Pang et al., 2020) introduces an aggregate interaction module to handle feature fusion challenges. (Yuan et al., 2020) proposes teacher-free knowledge distillation with improved label smoothing regularization. (Yang et al., 2021) introduces mutual learning by considering network width and resolution simultaneously. Inspired by these methods, we propose mutual distillation learning for PSIS. It achieves effective fusion and information promotion from instances and semantics, thereby improving the quality of segmentation results.

## 3 METHOD

**Overview.** As shown in Fig. 3, our proposed MDM consists of distillation from semantic to instance information and instance to semantic information. In our framework, HRNet serves as the backbone network for feature extraction, which following BESTIE, predicts offset maps, class maps, and instance affinity matrices separately. MDM is a recurrent learning module.

Initially, S2I branch obtains initial offset maps from the generated semantic segmentation map, serving as the initial instance segmentation labels. The offset maps predicted by HRNet connect with the semantic segmentation maps to produce new instance segmentation maps.

Then, the I2S branch uses the newly generated instance segmentation maps to compute the instance affinity matrix. A new semantic segmentation map is generated by multiplying the affinity matrix with the semantic scores (*i.e.*, class map) predicted from HRNet. The instance segmentation map generated at the final stage of MDM is used as pseudo labels to supervise the learning of the segmentor.

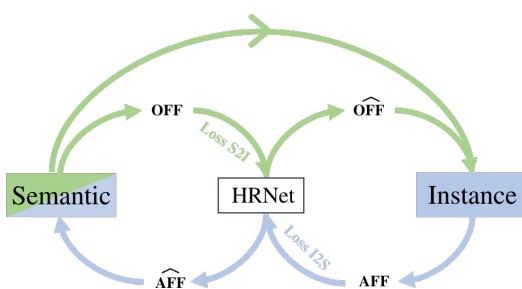

Figure 4: The abstract concept of semantic and instance information interaction. Corresponding to Fig. 3, the S2I branch is colored in green, while the I2S branch is colored in blue.

### 3.1 SEMANTIC TO INSTANCE

As shown in Fig. 5 regarding the information distilled from semantic to instance, we introduce the S2I branch that combines point annotations and weakly supervised semantic segmentation results to create initial instance segmentation labels. It also calculates instance-level offset maps by measuring inter-pixel distances within the initial results. Initially, it identifies preliminary instance regions using connected component labeling based on the semantic segmentation. Then, these regions are matched with the set of point annotations $E = \{e_1, e_2, \cdots, e_K\}$, where $K$ denotes the number of point annotations. In cases of conflicts (*e.g.*, multiple annotations in one region), a distance separation strategy is used. The distance is defined as:

$$k^* = \arg\min_k \|e_k - p\|, \tag{1}$$

where $p$ denotes the points in semantic segmentation map, $e_k$ denotes the $k$-th annotation point, $k^*$ represents the point $p$ assigned to the $k^*$-th annotation point $e_{k^*}$. After obtaining the initial

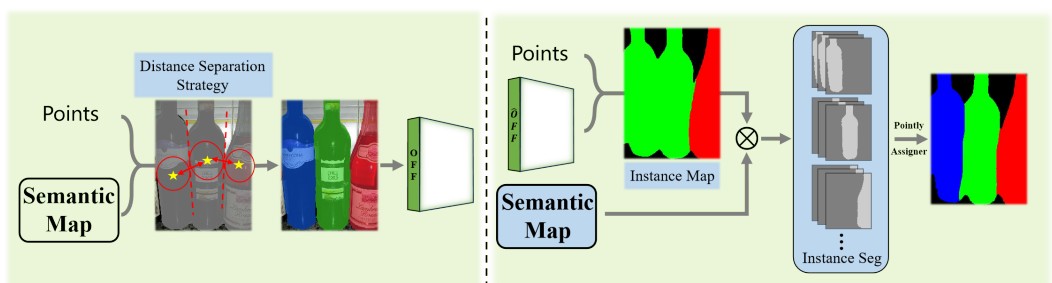

Figure 5: The architecture of the S2I branch. **Left:** The semantic segmentation map generates the offset map and class maps as supervised. **Right:** The process of generating the new instance segmentation map.

instance segmentation labels from the distance separation strategy, it generates the corresponding offset maps from the initial labels. Similar to BESTIE, we calculate the offset map for all instance points belonging to the annotation points by calculating the offset between the annotation and other points. The formula is as follows:

$$\mathcal{O} = \bigcup_{e_k \in E} \|e_k - m\|, \tag{2}$$

where $m$ denotes the points within the instance region to which the $k$-th annotated point $e_k$ belongs. $\mathcal{O}$ denotes the offset maps.

The offset maps generated by HRNet prediction is not only used for generating class-agnostic instance segmentation maps based on points but also contributes to the calculation of the loss function with initial offset map to facilitate further network learning through gradient back-propagation. The class-agnostic instance segmentation map is multiplied with the semantic segmentation map generated through I2S branch, leading to new instance segmentation results. It is important to note that instance adaptive grouping is based on point annotations for mask grouping. Our network creates final instance segmentation labels based on a combination of predicted semantic segmentation results, instance similarity matrices, and point annotations. In case of discrepancies between semantic and point-based segmentation, point annotations take precedence. For points without clear instance predictions, a default pseudo-box of size 16×16 is assigned as their segmentation prediction.

In the first stage, semantic map for S2I consists of pseudo-labels generated from the off-the-shelf segmentation map (*e.g.* SAM). Since the second stage, the semantic map input to MDM is generated from the previous stage. The instance segmentation map generated in the final stage will serve as pseudo-labels for the instance segmentor and perform the ultimate fine-tuning of the model.

## 3.2 INSTANCE TO SEMANTIC

As shown in the blue region in Fig. 3 , through the I2S branch, semantic information is distilled into instances. First, the branch obtains an affinity matrix from the instance segmentation map generated by S2I, representing the instance affinity between pairs of pixels. Specifically, in the instance similarity matrix, if two pixels belong to the same instance, their value is set to 1, otherwise, it is set to 0. Subsequently, this branch uses the affinity matrix $\mathcal{A} \in \mathbb{R}^{hw \times hw}$ predicted by HRNet as weights to adjust the class map $\mathcal{C} \in \mathbb{R}^{hw \times (c+1)}$ generated by HRNet. This adjustment enables the class map to incorporate instance information and generate the updated semantic segmentation map $\mathcal{S} = \mathcal{A}^{\circ\beta} * \mathcal{C}$, where $\mathcal{A}^{\circ\beta}$ denotes the $\beta$ times Hadamard power of $\mathcal{A}$, which smoothens the distribution to attain the optimal semantic segmentation map. Then affinity matrix generated from the instance segmentation map is used to compute the $\mathcal{L}_{I2S}$ (see Sec. 3.3) with the predicted affinity from HRNet to facilitate gradient backpropagation and promote self-learning of the network. The semantic segmentation map obtained through I2S serves as the semantic information input for the next stage. Through the constraints of instances, I2S enriches the semantic segmentation results with instance information. I2S can be widely applied to various weakly supervised segmentation tasks to allow the segmentation prediction with instance-wise information.

## 3.3 LOSS

As shown in Fig. 3, our framework consists of two branches corresponding to the respective two loss functions, namely the S2I loss $\mathcal{L}_{S2I}$, the I2S loss $\mathcal{L}_{I2S}$.

**S2I Loss**. S2I loss consists of the offset loss and the segmentation loss, *i.e.*, $\mathcal{L}_{S2I} = \mathcal{L}_{off} + \mathcal{L}_{seg}$, which is primarily used to constrain the network to capture instance information. Based on the annotations, we generate the offset map of the pixel value of the corresponding instance object according to the point annotations, which is represented by vector $(x, y)$. Offset map collects sets of pixels for labeled instance regions from pseudo labels, and set is denoted as $\mathcal{P}_{\text{pseudo}}$ . The offset loss is expressed as follows:

$$\mathcal{L}_{off} = \frac{1}{|\mathcal{P}_{\text{pseudo}}|} \sum_{(i,j) \in \mathcal{P}_{\text{pseudo}}} \mathcal{W}(i, j) \cdot \text{smooth}_{L1} |\hat{\mathcal{O}}(i, j) - \mathcal{O}(i, j)|, \qquad (3)$$

where predicted pseudos in the former stage are $\hat{\mathcal{O}}, \mathcal{O}$ , respectively. $\mathcal{W}$ means the weight of the pseudo loss. Offset loss represents the instance object's scale range that can well constrain the network regression. For semantic segmentation branch, the objective function is defined as:

$$\mathcal{L}_{seg} = \frac{1}{|\mathcal{P}_{seg}|} \sum_{(i,j) \in \mathcal{P}_{seg}} \text{CE}(p_{\hat{\mathcal{C}}(i,j)}, \mathcal{C}(i, j)), \qquad (4)$$

where $\hat{\mathcal{C}}, \mathcal{C}$ respectively represent the class map predicted by the network and the class map obtained at the S2I stage while acquiring the offset map, $p_{\hat{\mathcal{C}}(i,j)}$ represents the probabilities predicted by the network for each category in the class map, and $\mathcal{P}_{seg}$ is the set of all pixels in $\hat{\mathcal{C}}$.

**I2S Loss**. I2S loss uses the instance ownership relation between pixels to model the affinity matrix. The expression of I2S loss is as follows:

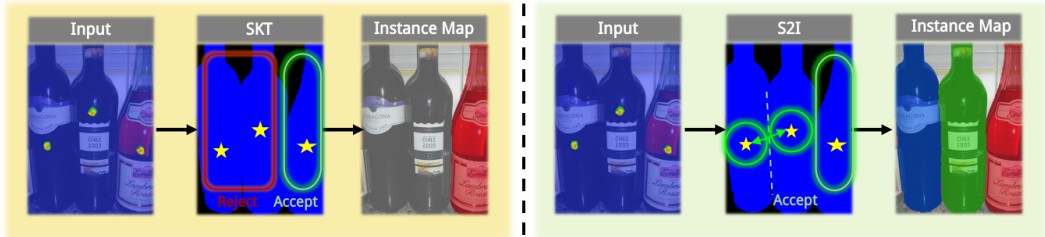

Figure 6: Comparison of distillation from S2I. The yellow area represents BESTIE method, and the green zone indicates the S2I branch.

$$\mathcal{L}_{I2S} = \frac{1}{N^+} \sum_{(i,j) \in R^+} \left( 2 - \sigma\left(\mathcal{A}^{ij}\right) - \sigma\left(\hat{\mathcal{A}}^{ij}\right) \right) + \frac{1}{N^-} \sum_{(k,l) \in R^-} \left( \sigma\left(\mathcal{A}^{kl}\right) + \sigma\left(\hat{\mathcal{A}}^{kl}\right) \right), \quad (5)$$

where $\sigma$ represents the sigmoid activation function. $R^+$ and $R^-$ denote the set of affinity and non-affinity pixel-pair samples. $(i, j)$, $(k, l)$ denote the pixel pairs, respectively. $N^+$ and $N^-$ count the number of $R^+$ and $R^-$, $\mathcal{A}$ and $\hat{\mathcal{A}}$ means the the affinity matrix generated from the instance map produced by the S2I process and predicted instance affinity matrix by the network HRNet. Eq. 5 reinforces the network to learn highly confident instance affinity relations from iterative training. On the other hand, since the instance affinity prediction is used by semantic segmentation branch, it also benefits the learning of global instance features and further helps to discover the integral object regions. The overall loss is finally formulated as $\mathcal{L} = \lambda_{I2S}\mathcal{L}_{I2S} + \lambda_{S2I}\mathcal{L}_{S2I}$.

## 4 EXPERIMENT

### 4.1 IMPLEMENTATION DETAILS

**Datasets.** Our experiments are conducted on PASCAL VOC 2012 (Everingham et al., 2010) and MS COCO 2017 (Lin et al., 2014) datasets. VOC dataset contains 20 instance classes, which is usually augmented with the SBD (Hariharan et al., 2011) dataset. It includes 10,582, 1,449, and 1,464 images for training, validation, and testing, respectively. COCO dataset contains 80 classes and includes $118k$ images for training and $5k$ images for validation. The images we used for training are annotated with point-level labels only.

**Evaluation Metrics.** We assess the performance of instance segmentation using two measures. We measure the performance using the standard protocol mean Average Precision(mAP). First, we measure the performance using the standard protocol mean Average Precision(mAP) with intersection-over-union(IoU) thresholds of 0.5, 0.7, and 0.75 for VOC. Second, the averaged AP over IoU thresholds from 0.5 to 0.95 is used on COCO dataset.

**Training details.** We use an AdamW (Loshchilov & Hutter, 2017) optimizer to train our network. The initial learning rate is set as $5 \times 10^{-5}$ and the batch size is set as 8. For the experiments on the VOC dataset, we train the network for 50,000 iterations. To ensure the initial pseudo labels are favorable, we warm-up the network for 2,000 iterations. For experiments on the COCO dataset, the total iterations number is 100,000 and the number of warm-up iterations is 5,000. The weights of semantic segmentation loss, offset map loss and instance affinity matrix loss are set as 1.0, 0.01, 1.0, respectively. The data augmentation strategies used here includes random resizing between 0.7 and 1.3, random cropping, random flipping, and photometric distortion. In the fine-tuning phase of the network's instance segmentor, we primarily employ the Mask R-CNN (He et al., 2017a) to retrain the instance segmentation map obtained from the last stage of MDM. We use SGD optimizer with a learning rate of 0.02 and the batch size is 16 with 12 epoches for both VOC and COCO dataset.

### 4.2 COMPARISON WITH STATE-OF-THE-ART METHODS.

**Results on Pascal VOC 2012.** We report the performance of the proposed method on the commonly used VOC validation dataset( $^\dagger$ represents the method reproduced in this paper). As shown in Table 1, the methods are grouped as image-supervised, box-supervised, and pointly-supervised methods from top to bottom. Our method achieves the best performance among all competing methods and even outperforms the most competitive BESTIE on both mAP$_{50}$ and mAP$_{75}$.

Table 1: Quantitative comparison of the state-of-the-art WSIS methods on VOC 2012 val-set. We denote the supervision sources as: $\mathcal{F}$ (full mask), $\mathcal{B}$ (box-level label), $\mathcal{I}$ (image-level label), $\mathcal{P}$ (point-level label), and $\mathcal{C}$ (object count). The off-the-shelf proposal techniques are denoted as follows: $\mathcal{M}$ (segment proposal (Pont-Tuset et al., 2016)), $\mathcal{R}$ (region proposal (Uijlings et al., 2013)), and $\mathcal{S}_{\mathcal{I}}$ (salient instance segmentor (Fan et al., 2019)).

| Method | Sup. | Backbone | Extra | mAP$_{25}$ | mAP$_{50}$ | mAP$_{70}$ | mAP$_{75}$ |
|---|---|---|---|---|---|---|---|
| Mask R-CNN (He et al., 2017a) | $\mathcal{F}$ | ResNet-50 | - | 76.7 | 67.9 | - | 44.9 |
| *End-to-End weakly-supervised models.* | | | | | | | |
| PRM (Zhou et al., 2018) | $\mathcal{I}$ | ResNet-50 | $\mathcal{M}$ | 44.3 | 26.8 | - | 9.0 |
| IAM (Zhu et al., 2019) | $\mathcal{I}$ | ResNet-50 | $\mathcal{M}$ | 45.9 | 28.3 | - | 11.9 |
| Label-PEnet (Ge et al., 2019) | $\mathcal{I}$ | VGG-16 | $\mathcal{R}$ | 49.2 | 30.2 | - | 12.9 |
| CL (Hwang et al., 2021) | $\mathcal{I}$ | ResNet-50 | $\mathcal{M},\mathcal{R}$ | 56.6 | 38.1 | - | 12.3 |
| BBTP (Hsu et al., 2019) | $\mathcal{B}$ | ResNet-101 | - | 23.1 | 54.1 | - | 17.1 |
| BBTP w/CRF | $\mathcal{B}$ | ResNet-101 | - | 27.5 | 59.1 | - | 21.9 |
| BoxInst (Tian et al., 2021) | $\mathcal{B}$ | ResNet-101 | - | - | 60.1 | - | 34.6 |
| OCIS (Cholakkal et al., 2019) | $\mathcal{C}$ | ResNet-50 | $\mathcal{M}$ | 48.5 | 30.2 | - | 14.4 |
| Point2Mask Li et al. (2023) | $\mathcal{P}$ | ResNet-101 | - | - | 48.4 | - | 22.8 |
| *Multi-Stage weakly-supervised models.* | | | | | | | |
| WISE (Laradji et al., 2019) | $\mathcal{I}$ | ResNet-50 | $\mathcal{M}$ | 49.2 | 41.7 | - | 23.7 |
| IRN (Ahn et al., 2019) | $\mathcal{I}$ | ResNet-50 | - | - | 46.7 | 23.5 | - |
| LIID (Liu et al., 2020) | $\mathcal{I}$ | ResNet-50 | $\mathcal{M},\mathcal{S}_{\mathcal{I}}$ | - | 48.4 | - | 24.9 |
| Arun *et al.* (Arun et al., 2020) | $\mathcal{I}$ | ResNet-101 | $\mathcal{M}$ | 59.7 | 50.9 | 30.2 | 28.5 |
| WISE-Net (Laradji et al., 2020) | $\mathcal{P}$ | ResNet-50 | $\mathcal{M}$ | 53.5 | 43.0 | - | 25.9 |
| BESTIE[†] (Kim et al., 2022) | $\mathcal{P}$ | HRNet-48 | - | 62.8 | 52.8 | - | 31.2 |
| SAM (Kirillov et al., 2023a) | $\mathcal{P}+\mathcal{S}$ | ViT-S/22.1M | - | 59.4 | 39.9 | - | 19.0 |
| **Ours** | $\mathcal{P}$ | ResNet-101 | - | **64.1** | **53.9** | **37.7** | **32.0** |
| **Ours** | $\mathcal{P}$ | HRNet-48 | - | **65.8** | **55.6** | **40.2** | **34.4** |

Table 2: Quantitative comparison of the state-of-the-art WSIS methods on MS COCO 2017 val-set. We denote the supervision sources as: $\mathcal{F}$ (full mask), $\mathcal{B}$(box-level label), $\mathcal{I}$(image-level label), and $\mathcal{P}$ (point-level label). The off-the-shelf proposal techniques are denoted as follows: $\mathcal{M}$ (segment proposal (Pont-Tuset et al., 2016)).

| Method | Sup. | Backbone | Extra | AP | AP$_{50}$ | AP$_{75}$ |
|---|---|---|---|---|---|---|
| Mask R-CNN (He et al., 2017a) | $\mathcal{F}$ | ResNet-50 | - | 34.6 | 56.5 | 36.6 |
| *End-to-End weakly-supervised models.* | | | | | | |
| BBTP (Hsu et al., 2019) | $\mathcal{B}$ | ResNet-101 | - | 21.1 | 45.5 | 17.2 |
| BoxInst (Tian et al., 2021) | $\mathcal{B}$ | ResNet-101 | - | 31.6 | 54.0 | 31.9 |
| Point2Mask (Li et al., 2023) | $\mathcal{P}$ | ResNet-101 | - | 12.8 | 26.3 | 11.2 |
| *Multi-Stage weakly-supervised models.* | | | | | | |
| IRN (Ahn et al., 2019) | $\mathcal{I}$ | ResNet-50 | - | 6.1 | 11.7 | 5.5 |
| WISE-Net (Laradji et al., 2020) | $\mathcal{P}$ | ResNet-50 | $\mathcal{M}$ | 7.8 | 18.2 | 8.8 |
| BESTIE[†] (Kim et al., 2022) | $\mathcal{P}$ | HRNet-48 | - | 14.2 | 28.4 | 22.5 |
| **Ours** | $\mathcal{P}$ | HRNet-48 | - | 17.6 | 33.6 | **28.1** |

**Results on MS COCO 2017.** The proposed P2Seg is compared with the state-of-the-art on more challenging COCO dataset( [†] represents the method reproduced in this paper). In Table 2, our method achieves the best performance among all competing methods and even outperforms the most competitive BESTIE on both AP$_{50}$ and AP$_{75}$. We find that P2Seg significantly outperforms weakly supervised methods and is on par with top-performing box-supervised methods.

**Overall Analysis.** In summary of all the above experimental results, P2Seg significantly enhances AP75, leading to overall segmentation performance improvement. The experiments illustrate that the mutual distillation method, leveraging instance and semantic information, can effectively estimate target scale and shape without requiring subsequent network fine-tuning. This optimization enhances handling of fine-grained segmentation details and resolves challenges with adjacent objects of the same class, ultimately improving segmentation performance.

## 4.3 ABLATION STUDY AND ANALYSIS

In order to further investigate the effectiveness of the instance-based and semantic-based mutual distillation algorithm, this section will conduct corresponding ablation experiments on the mutual

Table 3: Ablation study for our S2I and I2S, compared with BESTIE.

| S→I | I→S | mAP$_{50}$ |
|---|---|---|
| BESTIE | | 52.8 |
| S2I | | 53.2 |
| S2I | I2S | 55.7 |

Table 4: Ablation study for different Segmentor Backbones.

| Method | Segmentor | mAP$_{50}$ |
|---|---|---|
| BESTIE | Mask-RCNN | 52.8 |
| Ours | | 55.6 |
| BESTIE | SOLOv2 | 51.9 |
| Ours | | 54.1 |

Table 5: Ablation experiment to analyze the impact of hard pixel ratio.

| Hard pixel ratio | mAP$_{50}$ |
|---|---|
| 0.1 | 52.0 |
| 0.2 | 50.8 |
| 0.4 | 51.5 |
| 0.8 | 50.9 |

Table 6: The comparison of BESTIE and our P2Seg for IoU with the ground truth.

| Method | IoU > 50 | IoU >70 | IoU >90 | overall IoU |
|---|---|---|---|---|
| Semantic Results First | 5782 | 4939 | 2440 | 58.49 |
| Points First | **9544** | **7417** | **2558** | **66.57** |

distillation method. Unless otherwise specified, the datasets used in this section are from the PAS-CAL VOC 2012 dataset.

**Effect of S2I.** The motivation of S2I is to distill the semantic segmentation information into instance segmentation information. Based on the point annotation, S2I also shows its powerful ability to merge semantic segmentation results with point annotations. As shown in Fig. 6, the yellow part illustrates the Semantic Knowledge Transfer method proposed in BESTIE, which has a similar role to the S2I proposed in this paper. However, BESTIE does not fully utilize existing point annotations, whose design only considers scenarios where point annotations and foreground regions have no conflicts, leading to the loss of adjacent objects. In contrast, S2I assigns a unique instance region to represent each point's instance, ensuring better distinction and allocation of instances. As shown in Table.3, S2I obtains 1.4 mAP$_{50}$ performance improvement on PSIS tasks compared with the previous methods, showing the effectiveness of S2I statistically.

**Effect of I2S.** I2S module improves instance discrimination in the same class by using the instance affinity matrix. As shown in Table 3, I2S raises mAP$_{50}$ from 53.2 to 55.7. The Hadamard power of the affinity matrix provides additional information.

**Other Segmentors for Instance Segmentation.** As shown in the Table 4, we further evaluate instance segmentation networks trained with our pseudo labels, comparing them to SOLOv2 (Wang et al., 2020) and Mask R-CNN (He et al., 2017a), which are effective networks for single-stage and two-stage methods, respectively. The table displays results of Mask R-CNN and SOLOv2 trained with our pseudo labels, showing significant improvement over the previous state-of-the-art BESTIE. Our method outperforms BESTIE by 2.8 mAP$_{50}$ on Mask R-CNN and 2.2 mAP$_{50}$ on SOLOv2, highlighting the superiority of mutual distillation over one-way distillation.

**Hard pixel ratio.** As shown in Table 5, the term "hard pixel ratio" refers to the proportion of challenging samples used in loss computation. A lower proportion of hard pixels during training yields higher mAP$_{50}$, which is the validated result without a segmentor. However, reducing this proportion can lead to overfitting on easier examples and limit the model's ability to handle diverse scenarios, resulting in decreased generalization performance. We set the hard pixel ratio to 0.2.

**Prediction Mask Quality Comparison.** Considering that the post-processing in this approach involves fine-tuning the Mask RCNN network, this study further evaluates the quality of the predicted masks obtained before inputting them into the Mask RCNN network. Table 6 displays data from the PASCAL VOC 2012 dataset, showing the number of predicted masks with IoU thresholds of 0.5, 0.7, and 0.9 compared to ground truth masks in the first three columns. The "average IoU" column presents the mean IoU between predicted instance masks and ground truth masks.

**Backbone.** We make the ablation experiments about our method on the backbone layer of the network. As shown in Table 1, our method has consistent effects in different feature extracting methods from the backbone network.

### 4.4 VISUALIZATION

As shown in Fig. 7, the results demonstrate image segments using our method and the baseline. P2Seg effectively distinguishes boundaries between neighboring objects in the same class to enhance

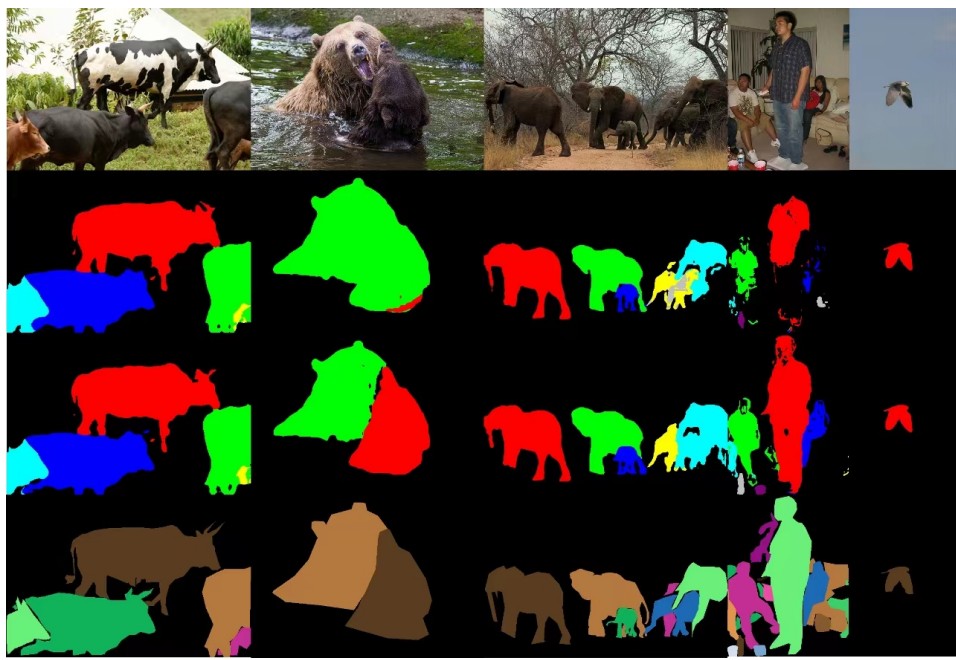

Figure 7: Visualization result on COCO. **First row**: Original image. **Second row**: Segmentation results of BESTIE before fine-tuning. **Third row**: Segmentation results of the point-supervised instance segmentation network constructed in this paper before fine-tuning. **Fourth row**: GT.

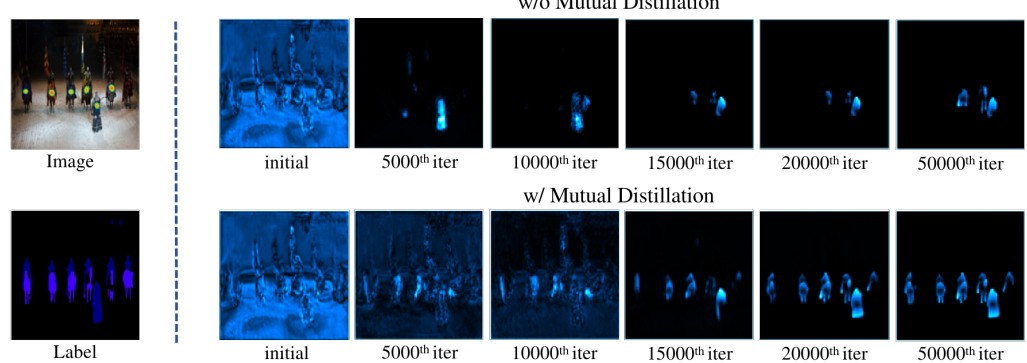

Figure 8: Comparison of class-agnostic instance segmentation learning. **Left**: Original image and class-agnostic instance segmentation annotations. **Right**: Learning progression from initialization to 50,000 iterations. The top without mutual distillation, and the bottom with it.

completeness and boundary clarity. The study validates the impact of mutual distillation on instance features through visualizations of class-agnostic instance segmentation maps. As shown in Fig. 8, we compare the segmentation results with and without mutual distillation reveals that the network gains more instance information over time with MDM, allowing it to segment more objects. MDM method significantly enhances the point-supervised instance segmentation network.

## 5 CONCLUSION

We propose MDM, a mutual distillation network using point-level data. It effectively combines point annotations with features and facilitates mutual distillation of information between instance-level and semantic-level information, and distinguishes object boundaries within the same class. MDM employs S2I and I2S processes, enhancing simple class and localization attentions for fully supervised instance segmentation training. Trained on Pascal VOC 2012, MDM shows outstanding performance in both instance and semantic segmentation tasks, showing the power of mutual distillation.

## 6 ACKNOWLEDGMENTS

This work was supported in part by the Youth Innovation Promotion Association CAS, the National Natural Science Foundation of China (NSFC) under Grant No. 61836012 and 61771447, the Strategic Priority Research Program of the Chinese Academy of Sciences under Grant No.XDA27000000.

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
