# A APPENDIX

## A.1 MORE EXPERIMENT RESULTS

**COCO Test Set Results**: Regarding the COCO test set results, our model achieved an AP of 17.4. We give the performance comparison in the Table.7.

Table 7: Quantitative comparison of the state-of-the-art WSIS methods on MS COCO 2017 test-dev. We denote the supervision sources as: $\mathcal{F}$ (full mask), $\mathcal{I}$(image-level label), and $\mathcal{P}$ (point-level label). The off-the-shelf proposal techniques are denoted as follows: $\mathcal{M}$ (segment proposal (Pont-Tuset et al., 2016))and$\mathcal{S}_\mathcal{I}$ (salient instance segmentor (Fan et al., 2019)).

| Method | *Sup.* | Extra | AP | $AP_{50}$ | $AP_{75}$ |
|---|---|---|---|---|---|
| | | *COCO test-dev.* | | | |
| Mask R-CNN (He et al., 2017a) | $\mathcal{F}$ | - | 35.7 | 58.0 | 37.8 |
| Fan *et al.* (Fan et al., 2018) | $\mathcal{I}$ | - | 13.7 | 25.5 | 13.5 |
| LIID (Liu et al., 2020) | $\mathcal{I}$ | $\mathcal{M},\mathcal{S}_\mathcal{I}$ | 16.0 | 27.1 | 16.5 |
| BESTIE[†] (Kim et al., 2022) | $\mathcal{P}$ | - | 14.2 | 28.6 | 12.7 |
| **Ours** | $\mathcal{P}$ | - | **17.4** | **33.3** | **16.4** |

**Quantitative analysis**: As indicated in Table.8, this analysis is grounded in two key metrics: Missing Rate and Adjacent Rate. The Missing Rate quantitatively evaluates the instances where our method fails to segment objects as precisely as depicted in the ground truth. It is noteworthy that, in comparison to the BESTIE method, our approach demonstrates a significant reduction in the occurrence of object misses, with a Miss Rate that is lower by 3.9%. On the other hand, the Adjacent Rate metric assesses the proportion of successfully identified adjacent objects. Here, our method markedly enhances the capability to recognize neighboring objects, exhibiting a 30.4% higher Adjacency Rate than that achieved by BESTIE.

**Influence of WSSS method**: The results in Table.9 shows how the WSSS result affects the WSIS. Originally, we adopt PMM (Li et al., 2021) for our WSSS method, which shows 70.0% mIoU and 55.6% $mAP_{50}$ on VOC 2012 validation set. Additionally, we train with ground-truth semantic segmentation labels and obtain a performance gain of 4.1% $mAP_{50}$; this result leaves us the opportunity that the advancement of the WSSS method can improve the performance of our approach.

**The parameter $\beta$**: As shown in Table.10, setting $\beta$ (the times of Hadamard power of $\mathcal{A}$) to 1 enables our model to attain an optimal equilibrium, culminating in the most effective performance within our weakly-supervised framework. It is noteworthy that an increment in $\beta$ corresponds to a decrement in performance by 0.5%.

**Analyze the efficiency**: We have conducted a comprehensive comparison of our method with the BESTIE method in terms of GFLOPs and FPS. The results of this comparison, as detailed in Table.11 as follows, reveal that our method only slightly increases the GFLOPs and FPS values compared to BESTIE. The increase in computation is mainly attributed to the additional label generator component of our method, which produces labels for training a fully supervised segmenter. However, during the inference stage, only the fully supervised segmenter is utilized, which means the inference time remains the same as that of BESTIE.

**Effect of I2S**: The I2S module improves instance discrimination in the same class by using the instance affinity matrix. As shown in Table.12, I2S raises $mAP_{50}$ from 17.4 to 17.6. This discrepancy in performance enhancement between the two datasets(Table.3 and Table.12) can likely be attributed to their inherent differences in features and complexity. The variations in performance improvements across datasets suggest that each module plays a crucial role, depending on the specific characteristics and challenges presented by the dataset.

**Drift-point**: As shown in Table.13, we conducted an ablation study where we applied Gaussian random perturbations with standard deviations ($\sigma$) of 5, 10, and 15 to the coordinates of the center points of each object. Our results reveal that when the centroid points are subject to a Gaussian perturbation with $\sigma = 5$, the scores are consistent with those obtained when the centroids are precisely located. However, as the standard deviation increases to $\sigma = 10$ and $\sigma = 15$, we observe a decrement in mAP50 by 0.6% and 1.5%, respectively, compared to the precise centroid placement. Our

findings indicate that while the performance of our method does experience a slight impact when the points undergo displacement, the overall effect remains within an acceptable range.

Table 8: Quantitative analysis for segmenting adjacent objects and addressing missing object issues.

| Method | Missing Rate | Adjacent Rate |
|---|---|---|
| BESTIE | 46.8 | 22.2 |
| **Ours** | **42.9** | **52.6** |

Table 9: Analysis of the effect of WSSS result on our WSIS performance.

| Semantic Segmentation | | Instance Segmentation |
|---|---|---|
| WSSS method | mIoU | $mAP_{50}$ |
| PMM | 70 | 55.6 |
| Ground Truth | - | 59.7 |

Table 10: Ablation experiment to analyze the impact of $\beta$.

| $\beta$ | $mAP_{50}$ |
|---|---|
| 1 | 51.5 |
| 2 | 51.0 |
| 3 | 51.0 |
| 4 | 51.0 |

Table 11: The ablation experiment to analyze the efficiency. Comparison of our method with the BESTIE method in terms of GFLOPs and FPS.

| Method | GFLOPS | FPS |
|---|---|---|
| BESTIE | 64.7 | 86.9 ms/img |
| **Ours** | **66.1** | **94.5 ms/img** |

Table 12: Ablation study for our S2I and I2S, compared with BESTIE on COCO.

| | S→I | I→S | $mAP_{50}$ |
|---|---|---|---|
| BESTIE | | | 14.2 |
| S2I | | | 17.4 |
| S2I | | I2S | 17.6 |

Table 13: Ablation experiment to analyze the impact of point drift. We apply Gaussian random perturbation to the coordinates of the center point of each object.

| Drift point($\sigma$) | $mAP_{50}$ |
|---|---|
| Center point | 64.4 |
| 5 | 64.4 |
| 10 | 63.8 |
| 15 | 62.9 |

## A.2    MORE VISUAL RESULTS

As shown in Fig. 9, Fig. 10, Fig. 11 and Fig. 12, the results of COCO and VOC demonstrate image segments using our method and the baseline. P2Seg effectively distinguishes boundaries between neighboring objects in the same class, enhancing completeness and boundary clarity. The study validates the impact of mutual distillation on instance features through visualizations of class-agnostic instance segmentation maps.

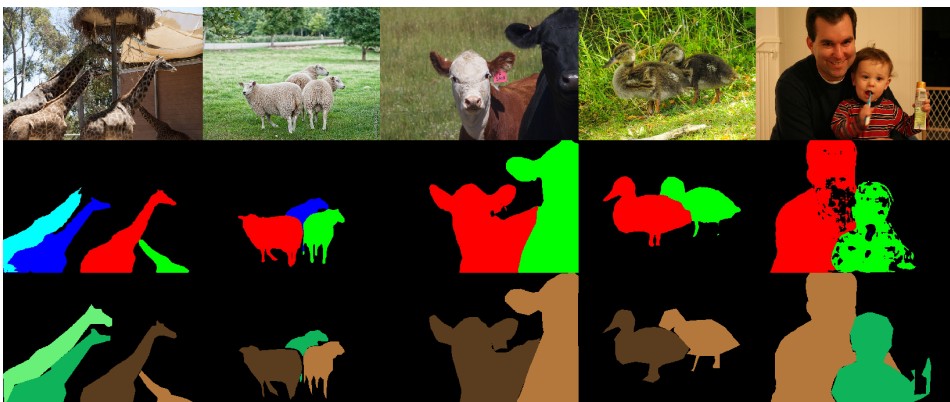

Figure 9: Visualization result comparison for our labels with ground truth on COCO. We can observe that our results are similar to the ground truth, indicating the effectiveness of the mutual distillation model we proposed.

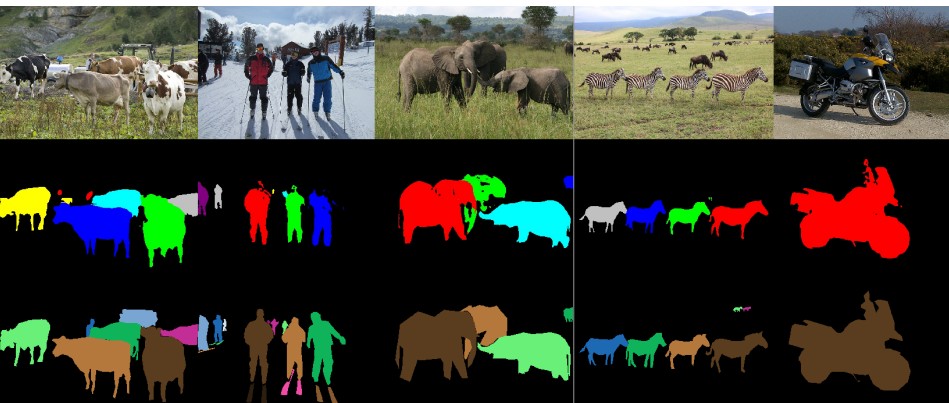

Figure 10: Visualization result comparison for our labels with ground truth on COCO.The first row is the original image. The second row presents the segmentation results of the point-supervised instance segmentation network constructed in this paper before inputting the instance segmentation results into the segmentor. The third row is the ground truth (GT) segmentation before .

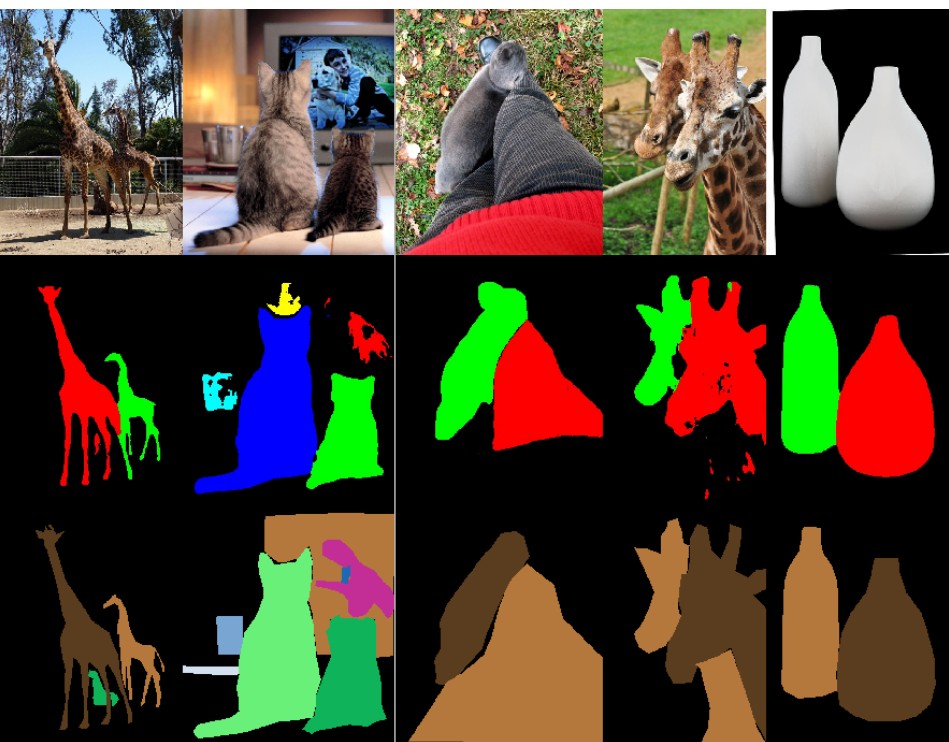

Figure 11: Visualization result comparison for our labels with ground truth on COCO. We can observe that our method is capable of effectively separating instances with overlapping edges.

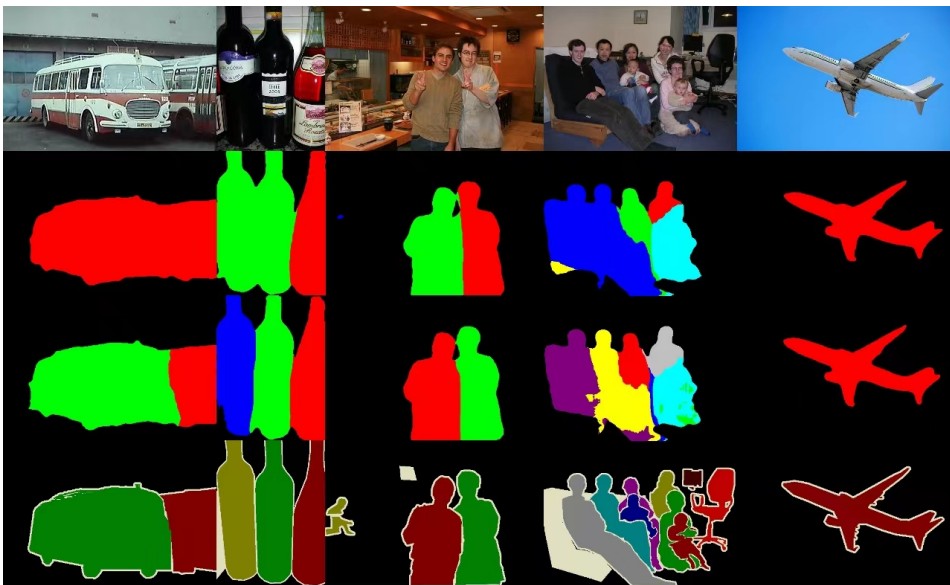

Figure 12: Visualization results comparison for our labels with ground truth on Pascal VOC 2012. The first row is the original image. The second row shows the segmentation results of BESTIE before fine-tuning. The third row presents the segmentation results of the point-supervised instance segmentation network constructed in this paper before fine-tuning. The fourth row is the ground truth (GT) segmentation.