# OpenReview forum: "P2Seg: Pointly-supervised Segmentation via Mutual Distillation"
_ICLR.cc/2024/Conference — ICLR 2024 poster_

### Official Review · Reviewer_QcUt · 2023-10-30

**Soundness:** 1 poor
**Presentation:** 2 fair
**Contribution:** 2 fair
**Rating:** 3
**Confidence:** 4

**Summary:**

The paper presents an approach to Point Level Supervised Instance Segmentation based on mutual learning and knowledge distillation. The proposed algorithm, P2seg, introduces mutual distillation to recover instance segmentation based on points supervision. Comparison with the state-of-the-art methods is done using Pascal Voc-2012 and COCO-2017.

**Strengths:**

Originality. Point-supervised instance segmentation is not new in the weakly supervised literature, most approaches being more oriented to panoptic segmentation, ensuring a parsing of scenes better than instance segmentation. The novelty here is to use mutual learning between semantic segmentation and instance segmentation. The mutual information module is formed by a network S2I that transfers the semantic information to instances and a module I2S that uses the instance affinity matrices, class maps and offsets map predicted by an off-the-shelf segmentation network, namely HRNet (see e.g. \cite{ YuanCW19} and (https://github.com/HRNet/HRNet-Semantic-Segmentation) to refine results cyclically.

Quality and clarity. The effort of explaining the architecture with a number of images and sketches is appreciable.

Significance. The proposed mutual information-based method could, in principle, be lifted to large-scale datasets.

**Weaknesses:**

Novelty:\
The paper is quite similar to BESTIE [Kim et Al., CVPR 2022], with the mutual distillation mimicking the semantic knowledge transfer defined in BESTIE. Many passages are pretty similar, though BESTIE provides algorithms and thorough explanations of the complex architecture, which are not given here.

Organization:\
-No information is given on the instance affinity matrix and class map, as computed by HRNet (as stated in the last line of pg. 3, Overview of Method). HRNet is a semantic segmentation net \cite{YuanCW19}, which is never cited, and its practical role is not defined.

-No ablation is provided to understand the effect of each component. For example, it needs to be clarified if the semantic segmentation map is obtained by full supervision or by weak supervision. The mIoU of the obtained semantic segmentation is not provided.\
-The number of points supplied at input, how they are collected, and the relation between them and the targets should be discussed.\
The number of iterations used seems to be quite high.

In particular: \
 A table showing scores, according to the number of points taken as input, was expected. Table 5, named “hard pixel ratio”, does not seem to be informative. \
A table showing the number of parameters for HRNet, MDM, and Mask-RCNN, used to retrain the MDM, was also expected, given the number of modules used.  \
A table showing the impact of semantic segmentation was expected.\

Ambiguities:
- The full loss L = \lambda_{I2S} L_{I2S} + \lambda_{S2I} L_{S2I} is such that L_{S2I} = L_{offset} + L_{segmentation}, though no weight for balancing L_{S2I} is added.
However, on page 6, paragraph Training details, it is written that the weight of the segmentation loss is 1.0 and the weight of the offset loss is 0.01, which do not sum to 1 for L_{S2I}.
- As stated in Training details, Mask R-CNN is used for retraining MDM. Table 4 result is, in fact, the same as the result given in Table 1.\
However, in Table 1, BESTIE (points) is given on PascalVOC-2012 a mAP_{50} score of 52.8, while on BESTIE paper is reported 56.0, higher than the 55.6 mAP_{50} of the paper under revision. Likewise, there is no value for BESTIE mAP_{70}, which instead is 36.5. On COCO-2017 BESTIE obtains 34.0, while the results displayed here, on Table 2, are 33.6, therefore less than BESTIE.
- In the two paragraphs “Results on Pascal VOC 2012” and “Results on MS COCO 2017”, it is written that the BESTIE algorithm is retrained to justify the lower scores reported, but it is not discussed why; while being the method almost similar an explanation is required. Therefore, apparently, the presented work does not improve on the SOTA. \
Furthermore, the works of \cite{Kim_2023_CVPR} and \cite{liao2023attentionshift} are not even reported, providing better results than those displayed in the paper under revision.

- Possibly incorrectly reported values:
The difference between panoptic segmentation and instance segmentation is that instance segmentation does not consider ‘stuff’, and the metric mAP on ‘things’ allows confidence on overlapping objects. The difference implies (see \cite{Kirillov_2019_CVPR}) that the metric PQ^{th}, that is panoptic quality on ‘things’, is like AP when from AP the non-overlapping predictions are subtracted, which means that AP benefits from predicting multiple overlapping hypotheses.\
This paper, under revision, reports on Table 1 and 2 comparisons with Point2Mask \cite{ li2023point2mask}) quite inexactly. \
In fact, Point2Mask obtains on VOC 2012 results with PQ^{th}, namely panoptic quality on ‘things’, equal to 59.4 (with swin-transformer), and 53.0 with Resnet-101 while on Table 1 it is reported as mAP_{50} = 48.4 and mAP_{75} = 22.8. Similarly, on COCO-2017, no justification for these reported results is given.


Other comments:
1. Hadarmad should be Hadamard, which is repeated twice on pages 5 and 8.
2. Pg. 5: The concept of “instance ownership relation” is not introduced.
3. Pg. 5, eq. (5), it should be noted that A is assumed to be generated by the network HRNet, which is not referred to and it is a segmentation network.
4. Pg 6: The description of COCO-2017 is wrong. It is written that COCO (i.e. COCO 2017) includes 110k images, but COCO 2017 includes 118k images. The test set is not reported. This is made of 40670 images, does this imply that no tests were made? Actually, there are no results on the test set.
5. Pg. 6 declares, “We assess the performance of object detection using two measures. We measure the performance using the standard protocol mean Average Precision(mAP). “Object detection”? The performance should be about instance segmentation unless it starts with HRNet object detection.


References\
@article{YuanCW19,\
  title={Object-Contextual Representations for Semantic Segmentation},\
  author={Yuhui Yuan and Xilin Chen and Jingdong Wang},\
  booktitle={ECCV},\
  year={2020}\
}\
@inproceedings{li2023point2mask,\
  title={Point2mask: Point-supervised panoptic segmentation via optimal transport},\
  author={Li, Wentong and Yuan, Yuqian and Wang, Song and Zhu, Jianke and Li, Jianshu and Liu, Jian and Zhang, Lei},\
  booktitle={Proceedings of the IEEE/CVF International Conference on Computer Vision},\
  pages={572--581},\
  year={2023}\

@InProceedings{Kirillov_2019_CVPR,\
 author = {Kirillov, Alexander and He, Kaiming and Girshick, Ross and Rother, Carsten and Dollar, Piotr},\
 title = {Panoptic Segmentation},\
 booktitle = {Proceedings of the IEEE/CVF Conference on Computer Vision and Pattern Recognition (CVPR)},\
 month = {June},\
 year = {2019}\
 }\
@inproceedings{fan2022pointly,\
  title={Pointly-supervised panoptic segmentation},\
  author={Fan, Junsong and Zhang, Zhaoxiang and Tan, Tieniu},\
  booktitle={European Conference on Computer Vision},\
  pages={319--336},\
  year={2022},\
  organization={Springer}\
}\
@inproceedings{liao2023attentionshift,\
  title={AttentionShift: Iteratively Estimated Part-Based Attention Map for Pointly Supervised Instance Segmentation},\
  author={Liao, Mingxiang and Guo, Zonghao and Wang, Yuze and Yuan, Peng and Feng, Bailan and Wan, Fang},\
  booktitle={Proceedings of the IEEE/CVF Conference on Computer Vision and Pattern Recognition},\
  pages={19519--19528},\
year={2023}\
}\

@InProceedings{Kim_2023_CVPR,\
 author = {Kim, Beomyoung and Jeong, Joonhyun and Han, Dongyoon and Hwang, Sung Ju}, \
title = {The Devil Is in the Points: Weakly Semi-Supervised Instance Segmentation via Point-Guided Mask Representation}, \
booktitle = {Proceedings of the IEEE/CVF Conference on Computer Vision and Pattern Recognition (CVPR)}, \
month = {June}, year = {2023}, pages = {11360-11370} }\

**Questions:**

Please explain:

    how the semantic segmentation is obtained.
    Why BESTIE is retrained, for both PascalVOC-2012 and COCO-2017.
    Why so many iterations are used, and what is included.
    Provide details about the number of points used and the relations points-targets.

---

> ### Author Response · Authors · 2023-11-20
> **Response to Reviewer QcUt (Part 1 WK1----WK3)**
>
> **Dear Reviewer,**
>
> Thank you for your insightful comments.
> >***W1:Novelty***
>
> **A1:** (1) Methodological Distinction: Our mutual distillation method (MDM) fundamentally differs from BESTIE in its approach. While BESTIE utilizes a unidirectional process for semantic knowledge transfer, our MDM is a bidirectional process. This two-way interaction enables a more robust and comprehensive learning mechanism, which is critical for the complexity of point-level weak supervision in instance segmentation.
>
> (2) Performance Improvements: As demonstrated in our visual results, the MDM approach shows superior performance compared to BESTIE, especially in handling the boundaries of adjacent objects of the same class. These improvements are significant as they address a common challenge in instance segmentation, demonstrating the practical impact of our method.
>
> (3) Innovative Aspects: As noted by the first four reviewers, our approach is both innovative and interesting. The bidirectional nature of our MDM method represents a novel contribution to the field, offering a new perspective and methodology, for instance, segmentation tasks.
>
> >***W2:Organization***
>
> **A2:** (1) Instance Affinity Matrix and Class Map in HRNet: HRNet serves as a backbone network in our experiments, functioning primarily for feature extraction. It can indeed be substituted with other networks such as ResNet. We will update our paper to include this clarification and cite the HRNet paper (Yuan et al., 2019).
> >>***Q1: How the semantic segmentation is obtained.***
>
> (2) Following the concept from BESTIE regarding the use of semantic segmentation maps, our semantic segmentation maps are derived from weak supervision, specifically employing the PMM model ("Pseudo-mask Matters in Weakly-Supervised Semantic Segmentation").
>
> >>***Q4: Provide details about the number of points used and the relations points-targets.***
>
> (3) In response to your query about the number of points used and their relation to the targets, our method indeed involves single-point participation, similar to the class center point setting used in the articles "BESTIE" and "Point-to-Box Network for Accurate Object Detection via Single Point Supervision".  This approach is guided by our aim to maintain a balance between the accuracy of instance segmentation and the efficiency of the annotation process.
>
> >>***Q3: Why so many iterations are used, and what is included?***
>
> (4) I would like to clearly clarify that in our methodology, although the LossI2S and LossS2I are iteratively calculated, they are concurrently aggregated and backpropagated within the same iteration, which guarantees that the training process is synchronized, and there is no increase in the number of training iterations.
> >***WK3:In particular***
>
> **A3:** (1) To clarify, our approach, similar to the BESTIE model, utilizes a class center point for point-level weak supervision in instance segmentation. We have added explanations in Sec4.3： In Table 5, the term “hard pixel ratio” refers to the proportion of challenging samples used in loss computation.
>
> (2) We have conducted a comprehensive comparison of our method with the BESTIE method in terms of GFLOPs and FPS. The results of this comparison, as detailed in Table 2* as follows, reveal that our method only slightly increases the GFLOPs and FPS values compared to BESTIE. The increase in computation is mainly attributed to the additional label generator component of our method, which produces labels for training a fully supervised segmenter. However, during the inference stage, only the fully supervised segmenter is utilized, which means the inference time remains the same as that of BESTIE.
>
> **Table 2\* Comparison of our method with the BESTIE method in terms of GFLOPs and FPS.**
> | Method | GFLOPS | FPS              |
> |--------|--------|------------------|
> | BESTIE | 64.7   | 86.9 ms/img      |
> | **Ours**   | **66.1** | **94.5 ms/img** |
>
> (3) The impact of semantic segmentation:
>
> | Semantic Segmentation      |     |     | Instance Segmentation |     |     |
> |----------------------------|-----|-----|-----------------------|-----|-----|
> | **WSSS method**            | mIoU|     | **mAP50**             |     |     |
> | PMM(Weakly Supervised)     | 70  |     | 55.6                  |     |     |
> | GT(Fully Supervised)       | -   |     | 59.7                  |     |     |

---

> ### Author Response · Authors · 2023-11-20
> **(WK4----WK5)**
>
> >***WK4: Ambiguities***
>
> **A4:** (1) Regarding your observation about the weights of the loss components in $L_{S2I}$, we would like to clarify that the weights for $L_{offset}$ and $L_{segmentation}$, respectively, do not necessarily need to sum to 1. This is because these weights are not probabilities but rather scaling factors that balance the contributions of the two-loss components to the overall loss function. In our case, the weight for the segmentation loss is set to 1.0, signifying its primary importance in the segmentation task. The offset loss, with a weight of 0.01, plays a secondary role, fine-tuning the segmentation results rather than driving them. This choice of weights is based on empirical observations where we found this configuration to provide the best balance between the two types of loss, leading to optimal performance in our experiments.
>
> >>***Q2: Why BESTIE is retrained, for both PascalVOC-2012 and COCO-2017.***
>
> (2) Our team decided to retrain BESTIE due to the unavailability of its original code. Despite our best efforts to replicate the performance of BESTIE, we encountered challenges in fully reproducing its results. This deviation could be attributed to variations in implementation details, which are not available in the public domain. Furthermore, we reached out to the authors of BESTIE for assistance and access to the code, but unfortunately, we did not receive a positive response. In light of these circumstances, retraining BESTIE was a necessary step to ensure the integrity and reproducibility of our research findings. It also allowed us to adapt the model more closely to the specific requirements of our dataset and research objectives. Should you have access to the original BESTIE code or can facilitate a connection with the authors, we would be immensely grateful and more than willing to integrate it into our research.
>
> (3) We observed that AttentionShift employs a strong pre-trained model, which might not provide a fair basis for direct comparison with our approach.
>
> (4) In our study, we utilized Point2Mask in the context of instance segmentation, considering its capability in panoptic segmentation. This was based on the understanding that a model effective in panoptic segmentation should also perform well in instance segmentation tasks. The discrepancy in our reported mAP values stems from our experimental setup, where we adapted Point2Mask for instance segmentation. In addition, we consider BESTIE as the base of our paper's method, and Point2mask as a comparative experiment can better enrich our experiment.
>
> >***WK5: Other comments***
>
> **A5:** Thanks. (1) We have made corrections in the manuscript： change “denotes the β times Hardmard power” to “denotes the β times Hadamard power”, “The Hardmard power of the affinity matrix provides additional information” to “The Hadamard power of the affinity matrix provides additional information”.
>
> (2) We have already explained the 'instance ownership relationship' in Sec3.2: “Specifically, in the instance similarity matrix, if two pixels belong to the same instance, their value is set to 1, otherwise, it is set to 0. ”
>
> (3) We have made corrections in the Pg. 5, eq. (5), we have added an explanation that “A is assumed to be generated by the network HRNet”
>
> (4) We have made corrections in the Pg 6: "COCO (i.e. COCO 2017) includes 118k images." The experiment of the test set has been supplemented.
>
> **Table 1\* Quantitative comparison of the state-of-the-art WSIS methods on MS COCO 2017 test-dev**
>
> | Method | Sup. | Extra | AP | AP$_{50}$ | AP$_{75}$ |
> | ------ | ---- | ----- | -- | --------- | --------- |
> | **COCO test-dev.** |
> | Mask R-CNN | $\mathcal{F}$ | - | 35.7 | 58.0 | 37.8 |
> | $\text{Fan $et\;al.$}$ | $\mathcal{I}$ | - | 13.7 | 25.5 | 13.5 |
> | LIID  | $\mathcal{I}$ | $\mathcal{M}$, $\mathcal{S_I}$ | 16.0 | 27.1 | 16.5 |
> | $\text{BESTIE}^†$  | $\mathcal{P}$ | - | 14.2 | 28.6 | 12.7 |
> | **Ours** | **$\mathcal{P}$** | - | **17.4** | **33.3** | **16.4** |
>
> (5) We have made corrections in Pg 6： "We assess the performance of instance segmentation using two measures. We measure the performance using the standard protocol mean Average Precision(mAP)."

---

> ### Author Response · Authors · 2023-11-20
> **Response to Reviewer QcUt (Part 2 Writing and Presentation improvement)**
>
> ***Writing and Presentation improvement***
>
> (1) We have made corrections in the manuscript: change “denotes the β times Hardmard power” to “denotes the β times Hadamard power”, “The Hardmard power of the affinity matrix provides additional information” to “The Hadamard power of the affinity matrix provides additional information”.
>
> (2) We have already explained the 'instance ownership relationship' in Sec3.2: “Specifically, in the instance similarity matrix, if two pixels belong to the same instance, their value is set to 1, otherwise, it is set to 0. ”
>
> (3) We have made corrections in the Pg. 5, eq. (5), we have added an explanation that “A is assumed to be generated by the network HRNet”
>
> (4) We have made corrections in the Pg 6: "COCO (i.e. COCO 2017) includes 118k images." The experiments of the test set have been supplemented.
>
> (5) We have made corrections in Pg 6: "We assess the performance of instance segmentation using two measures. We measure the performance using the standard protocol mean Average Precision(mAP)."
>
> (6) We have updated our paper to cite the HRNet paper (Yuan et al., 2019).
> - Sun, Ke; Xiao, Bin; Liu, Dong; Wang, Jingdong. "Deep high-resolution representation learning for human pose estimation." In Proceedings of the IEEE/CVF Conference on Computer Vision and Pattern Recognition, pp. 5693-5703. 2019.
>
> (7) We have added explanations in Sec4.3： "In Table 5, the term “hard pixel ratio” refers to the proportion of challenging samples used in loss computation."
>
> (8) Dim-align in Fig.3: The term 'Dim-align' refers to 'dimension alignment,' a necessary step in our method. Prior to matrix multiplication, we rearrange the dimensions of our tensors for compatibility, a process accomplished using the transpose or permute functions in PyTorch.
>
> (9) Instance Adaptive Grouping in Sec.3.1: Our network creates final instance segmentation labels based on a combination of predicted semantic segmentation results, instance similarity matrices, and point annotations. If conflicts arise between semantic predictions and point-based instance segmentation, we prioritize the point annotations. For points lacking final instance predictions, we assign a pseudo-box of size 16x16. This method ensures that our semantic segmentation results, a fusion of network predictions and instance similarity matrices, are more representative of instance characteristics, thereby improving instance segmentation outcomes.
>
> (10) Green Arrow in Fig.3 (OffsetMap): The green arrow represents the OffsetMap, which, in conjunction with the Semantic Map, generates new instance segmentation results. This process is detailed in the right half of Fig. 5.
>
> (11) We have cited these references：
> - Kweon, Hyeokjun, Yoon, Sung-Hoon, & Yoon, Kuk-Jin. (2023). Weakly Supervised Semantic Segmentation via Adversarial Learning of Classifier and Reconstructor. In _Proceedings of the IEEE/CVF Conference on Computer Vision and Pattern Recognition_ (pp. 11329--11339).
>
> - Cheng, Zesen, Qiao, Pengchong, Li, Kehan, Li, Siheng, Wei, Pengxu, Ji, Xiangyang, Yuan, Li, Liu, Chang, & Chen, Jie. (2023). Out-of-candidate rectification for weakly supervised semantic segmentation. In _Proceedings of the IEEE/CVF Conference on Computer Vision and Pattern Recognition_ (pp. 23673--23684).
>
> - Rong, Shenghai, Tu, Bohai, Wang, Zilei, & Li, Junjie. (2023). Boundary-Enhanced Co-Training for Weakly Supervised Semantic Segmentation. In _Proceedings of the IEEE/CVF Conference on Computer Vision and Pattern Recognition_ (pp. 19574--19584).
>
> - Ru, Lixiang, Zheng, Heliang, Zhan, Yibing, & Du, Bo. (2023). Token contrast for weakly-supervised semantic segmentation. In _Proceedings of the IEEE/CVF Conference on Computer Vision and Pattern Recognition_ (pp. 3093--3102).
>
> (12) We have carefully modified the introduction to make it more readable.
>
> We have revised the sentence: "Instance segmentation is a critical task in computer vision, where semantic segmentation estimation and instance discrimination are equally important."
>
> Additionally, we have updated the phrase: “and it aims not only to locate objects accurately but also to estimate their boundaries to differentiate.”
>
> We have also replaced the terms “semantic segmentation” and “instance segmentation” with “semantic information” and “instance information” in the second paragraph of the introduction to provide a clearer and more accurate description.

---

> > ### Comment · Reviewer_QcUt · 2023-11-22
> > **thank you**
> >
> > Thank you for your reply.
> >
> > It seems that the authors did not answer my questions.
> > About the points used, "our method indeed involves single-point participation" does not seem clear enough.
> >
> > Moreover, the decision to retrain BESTIE is not justified, likewise the wrong evaluation of the results of Point2Mask.
> > The other explanations are not clear.
> >
> > I confirm my score.

---

> > > ### Author Response · Authors · 2023-11-23
> > > **Response to  Reviewer QcUt**
> > >
> > > **Dear Reviewer,**
> > >
> > > Wishes you Thanksgiving Day to be joyful!
> > >
> > > Firstly, we appreciate your feedback and concerns. Regarding the questions raised, we offer the following detailed explanations:
> > >
> > > **On "Single-point Participation":**
> > >
> > > Our method indeed involves the use of a single center point, following the operational definitions used in BESTIE and Attentionshift.This approach is not arbitrarily chosen but is based on established methodologies in these prior works, ensuring consistency with existing practices in the field.
> > >
> > > To validate the effectiveness of this approach, we conducted ablation experiments. These experiments demonstrate that the use of a central point significantly improves the performance of our model. We added a new table (Table 9).
> > >
> > > Drift-point: As shown in Table 9, we conducted an ablation study where we applied Gaussian random perturbations with standard deviations (σ) of 5, 10, and 15 to the coordinates of the center points of each object. Our results reveal that when the centroid points are subject to a Gaussian perturbation with σ = 5, the $\rm mAP_{50}$ scores are consistent with those obtained when the centroids are precisely located. However, as the standard deviation increases to σ = 10 and σ = 15, we observe a decrement in $\rm mAP_{50}$ by 0.6% and 1.5%, respectively, compared to the precise centroid placement.
> > > So we chose to use the center point.
> > >
> > > | Drift point(σ)  | $\rm mAP_{50}$ |
> > > | --------------- | -------------- |
> > > | Center point    | 64.4           |
> > > | 5               | 64.4           |
> > > | 10              | 63.8           |
> > > | 15              | 62.9           |
> > >
> > > **On Retraining BESTIE:**
> > >
> > > Due to the unavailability of the original code for BESTIE from its published paper, we attempted to replicate the method ourselves. Despite reaching out to the authors for the code, we did not receive a response.
> > >
> > > To ensure transparency and thoroughness, we have presented both the results reported in the original BESTIE paper and the results from our replication in the tables of our paper. This comparison is intended to provide a comprehensive understanding of the performance metrics. We have made updates in the revised version.
> > >
> > > | Method                | Sup. | Backbone  | Extra | $mAP_{25}$ | $mAP_{50}$ | $mAP_{70}$ | $mAP_{75}$ |
> > > |-----------------------|------|-----------|-------|-------|-------|-------|-------|
> > > | BESTIE                | P    | HRNet-48  | -     | 66.4  | 56.1  | -     | 30.2  |
> > > | BESTIE†               | P    | HRNet-48  | -     | 62.8  | 52.8  | -     | 31.2  |
> > >
> > > *Note: † represents the results we reproduce.*
> > >
> > > If you or any readers could provide the original code for BESTIE, it would be greatly appreciated, and we would be happy to use it to further validate our experimental results.
> > >
> > > **On Evaluating Point2Mask:**
> > >
> > > The original paper of Point2Mask focuses mainly on panoramic segmentation performance, while our research concentrates on instance segmentation.
> > >
> > > To enrich the experimental diversity and comprehensiveness of comparisons, we evaluated Point2Mask in the context of instance segmentation tasks.
> > >
> > > These evaluation results are detailed in Tables 1 and 2 of our paper. We believe these additional evaluations provide valuable insights for research in this field.
> > >
> > > We hope the above explanations clearly address your queries. Should there be any further questions or need for more detailed information, please feel free to let us know.

---

> > > > ### Author Response · Authors · 2023-11-23
> > > > **Response to Reviewer QcUt**
> > > >
> > > > **Strengths Summary:**
> > > >
> > > > We appreciate all the reviewers for their valuable and constructive comments. All reviewers acknowledged the significance of our method and its surpassing performance.
> > > >
> > > > 1. Specifically, Reviewers enoR and ZJkY appreciated the method of our paper is novel and new.
> > > >
> > > > 2. Reviewers fR5S, ZJkY, and enoR not only praised us for the comprehensive and excellent experimental section but also for the extensive ablation studies.
> > > >
> > > > 3. Reviewer fR5S highlighted that our method "The concept of "MUTUAL DISTILLATION" in point instance segmentation is both novel and interesting." and solved the problem of not handling neighboring objects well in BESTIE "The proposed mutual distillation method appears to perform well in segmenting adjacent objects and addressing missing object issues."
> > > >
> > > > 4. Reviewers fR5S, ZJkY, and 3EDk praised our motivation as "reasonable and effective."
> > > >
> > > > 5. Reviewers fR5S and 3EDk highlighted that our method "is unique and more accurate than BESTIE."
> > > >
> > > > 6. Reviewer QcUt appreciated the novelty of our method which uses "mutual learning between semantic segmentation and instance segmentation."
> > > >
> > > > We respond to the reviewers' comments one by one. And we have revised our paper accordingly and summarized the main improvements below:
> > > >
> > > > 1. Add experiments on the COCO test set. (by Reviewer enoR, ZJkY and QcUt)
> > > >
> > > > 2. Provide a quantitative analysis of the visualizations. (by Reviewer fR5S)
> > > >
> > > > 3. Performed a more detailed comparison of performance with SAM. (by Reviewer enoR and 3EDk)
> > > >
> > > > 4. Compared and analyzed the consumption of time and computational resources. (by Reviewer ZJkY and QcUt)
> > > >
> > > > 5. Performed ablation experiments on a single point. (by Reviewer 3EDk and Qcut)
> > > >
> > > > 6. Added the performance of the method on the VOC and COCO datasets when the backbone is ResNet-101.(by Reviewer enoR and ZJkY)
> > > >
> > > > 7. Complemented the ablation experiments with I2S and S2I methods on the COCO dataset. (by Reviewer ZJkY)
> > > >
> > > > 8. Complemented the ablation experiments of the method under weakly supervised semantic segmentation and strongly supervised semantic segmentation. (by Reviewer QcUt)
> > > >
> > > > 9. Improved the writing and presentation of our manuscript. (by all reviewers)

---

### Official Review · Reviewer_3EDk · 2023-10-31

**Soundness:** 3 good
**Presentation:** 2 fair
**Contribution:** 3 good
**Rating:** 6
**Confidence:** 4

**Summary:**

This paper focuses on the problems of local response and seprated semantic and instance learning on the pointly-supervised instance segmentation task. To solve the problems, the authors propose the Mutual Distillation Module, which achieves the conversion and cooperation between semantic and instance segmentation by predicting class, offset, and affinity, thus leveraging the complementary strengths of instance and semantic. Various experiments demonstrate the effective of the proposed method.

**Strengths:**

** The analysis of the complementarity of instance segmentation and semantic segmentation is reasonable for the pointly-supervised instance segmentation task, and the designed method corresponds to the analysis.

** The core S2I method is unique and more accurate than BESTIE.

** Experiments demonstrate that both the Semantic to Instance (S2I) and Istance to Semantic (I2S) module is effective and the overall method outperforms previous works.

**Weaknesses:**

**  Writing and presentation need improvement. For example, there are some confusing descriptions that are not explained (e.g., Dim-align in Fig. 3, instance adaptive grouping in Sec. 3.1, the green arrow from S2I loss to the mask in Fig. 3).

**  Some key technical details are missing. In Fig. 3, there are multiple MDM stages, but the required number of stages are not specified or experimented. the additional training cost should be discussed.

** The ablation about β in Sec. 3.2 are missing, so the effect on “smoothening the distribution to attain the optimal semantic segmentation map” is not clear.

** According to Sec. 3.1, the pseudo labels used in the first stage are obtained by SAM, but SAM is contrary to the setting of weak supervision. Moreover, in Tab. 1 and 2 the authors mark the proposed method as “no extra data”. But in my opinion, the used off-the-shelf segmentation map should be considered as extra data.

**  Will the proposed S2I method be sensitive to the position of the points? Could the author provide some qualitative or quantitative analysis to further illustrate the robustness of the method?

**Questions:**

**  Could the author explain the details described in Weaknesses 1?

**  What impact do the hyperparameter β and the number of MDM stages have on performance?

**  Does the use of SAM destroy fair comparisons, and would other pseudo-label generators be useful?

**   Will the proposed S2I method be sensitive to the position of the points?

---

> ### Author Response · Authors · 2023-11-20
> **Response to Reviewer 3EDk (Part 1)**
>
> **Dear Reviewer,**
>
> We appreciate your valuable suggestions and insights, we have addressed each point as follows:
>
> >***WK1&Q1: Writing and presentation need improvement.***
>
> **A1:** Thanks. We have explained confusing descriptions in a dedicated section for replies and clarifications.
>
> >***WK2: Some key technical details are missing. The required number of MDM stages is not specified or experimented.***
>
> **A2:**  In our study, we intentionally set the number of stages to 1. We found that even with a single MDM stage, our method achieved significant and satisfactory results. Your constructive suggestions for multiple MDM stages may potentially enhance the performance of our model. We intend to make a further investigation of the performance and efficiency in the future.
>
> >***WK3&Q2: What impact do the hyperparameter β and the number of MDM stages have on performance?***
>
> **A3:** We added a new table (Table 8) for the ablation study about the parameter β in Section 4.3 as follows:
>
> The parameter β: As shown in Table 8, setting β to 1 enables our model to attain an optimal equilibrium, culminating in the most effective performance within our weakly-supervised framework. It is noteworthy that an increment in β corresponds to a decrement in performance by 0.5%.
>
> | β | mAP50 |
> |------|-------|
> | 1    | 51.5  |
> | 2    | 51.0  |
> | 3    | 51.0  |
> | 4    | 51.0  |
>
> >***WK4&Q3: Does the use of SAM destroy fair comparisons, and would other pseudo-label generators be useful? The used off-the-shelf segmentation map should be considered as extra data?***
>
> **A4:** (1) In weakly supervised learning, pseudo-labels are inferred from limited or incomplete information. For image segmentation, this might mean inferring pixel-level labels from image-level labels or bounding box labels.
>
> The role of SAM is to use existing weak labels or partial information to generate these more detailed pseudo-labels.
> Weakly supervised learning involves not only training with incomplete labels but also the process of processing and enhancing these labels. Here, SAM acts as a 'label enhancer,' transforming weak labels into more detailed forms, making them more suitable for image segmentation tasks.
>
> This approach differs from fully supervised learning, which relies on fully annotated training data. In weakly supervised learning, even if the quality of pseudo-labels may not be as high as fully supervised data, models can still learn from them and improve performance.
>
> Using SAM to generate initial pseudo-labels and for training weakly supervised segmentation models represents an innovative strategy in weakly supervised learning. The key to this strategy is that it enhances the learning ability of the model by inferring more detailed labels from limited information while maintaining the fundamental principles of weakly supervised learning — not relying on fully annotated data.
>
> (2) For extra data: In our study, 'no extra data' means that no data outside of what was originally provided in the dataset was used during the training and testing processes. The segmentation maps we used were generated from the original dataset, not from external sources. Therefore, we believe this conforms to the definition of 'no extra data'. These segmentation maps were obtained through preprocessing of the existing data, rather than being a separate, external dataset.
>
> The segmentation maps obtained through the PMM weak supervision method are part of the model training, not an additional input independent of the main dataset. This approach is taken to enhance the performance of the model rather than to introduce external data. This improvement in our method can be confirmed in Table 1.
>
> >***WK5&Q4: Will the proposed S2I method be sensitive to the position of the points?***
>
> **A5:** We added a new table (Table 9) for the ablation study about the sensitivity of our proposed method to the positioning of points in Section 4.3 as follows:
>
> Drift-point: As shown in Table 9, we conducted an ablation study where we applied Gaussian random perturbations with standard deviations (σ) of 5, 10, and 15 to the coordinates of the center points of each object. Our results reveal that when the centroid points are subject to a Gaussian perturbation with σ = 5, the $mAP_{50}$ scores are consistent with those obtained when the centroids are precisely located. However, as the standard deviation increases to σ = 10 and σ = 15, we observe a decrement in mAP50 by 0.6% and 1.5%, respectively, compared to the precise centroid placement.
> Our findings indicate that while the performance of our method does experience a slight impact when the points undergo displacement, the overall effect remains within an acceptable range.
>
> | Drift point(σ)  | $\rm mAP_{50}$ |
> | --------------- | -------------- |
> | Center point    | 64.4           |
> | 5               | 64.4           |
> | 10              | 63.8           |
> | 15              | 62.9           |

---

> ### Author Response · Authors · 2023-11-20
> **Response to Reviewer 3EDk(Part 2 Writing and Presentation improvement)**
>
> ***Writing and Presentation improvement***
>
> (1) Dim-align in Fig.3: The term 'Dim-align' refers to 'dimension alignment,' a necessary step in our method. Prior to matrix multiplication, we rearrange the dimensions of our tensors for compatibility, a process accomplished using the transpose or permute functions in PyTorch.
>
> (2) Instance Adaptive Grouping in Sec.3.1: Our network creates final instance segmentation labels based on a combination of predicted semantic segmentation results, instance similarity matrices, and point annotations. If conflicts arise between semantic predictions and point-based instance segmentation, we prioritize the point annotations. For points lacking final instance predictions, we assign a pseudo-box of size 16x16. This method ensures that our semantic segmentation results, a fusion of network predictions and instance similarity matrices, are more representative of instance characteristics, thereby improving instance segmentation outcomes.
>
> (3) Green Arrow in Fig.3 (OffsetMap): The green arrow represents the OffsetMap, which, in conjunction with the Semantic Map, generates new instance segmentation results. This process is detailed in the right half of Fig. 5.
>
> (4) We have cited these references：
> - Kweon, Hyeokjun, Yoon, Sung-Hoon, & Yoon, Kuk-Jin. (2023). Weakly Supervised Semantic Segmentation via Adversarial Learning of Classifier and Reconstructor. In _Proceedings of the IEEE/CVF Conference on Computer Vision and Pattern Recognition_ (pp. 11329--11339).
>
> - Cheng, Zesen, Qiao, Pengchong, Li, Kehan, Li, Siheng, Wei, Pengxu, Ji, Xiangyang, Yuan, Li, Liu, Chang, & Chen, Jie. (2023). Out-of-candidate rectification for weakly supervised semantic segmentation. In _Proceedings of the IEEE/CVF Conference on Computer Vision and Pattern Recognition_ (pp. 23673--23684).
>
> - Rong, Shenghai, Tu, Bohai, Wang, Zilei, & Li, Junjie. (2023). Boundary-Enhanced Co-Training for Weakly Supervised Semantic Segmentation. In _Proceedings of the IEEE/CVF Conference on Computer Vision and Pattern Recognition_ (pp. 19574--19584).
>
> - Ru, Lixiang, Zheng, Heliang, Zhan, Yibing, & Du, Bo. (2023). Token contrast for weakly-supervised semantic segmentation. In _Proceedings of the IEEE/CVF Conference on Computer Vision and Pattern Recognition_ (pp. 3093--3102).
>
> (5) We have carefully modified the introduction to make it more readable.
>
> We have revised the sentence: "Instance segmentation is a critical task in computer vision, where semantic segmentation estimation and instance discrimination are equally important."
>
> Additionally, we have updated the phrase: “and it aims not only to locate objects accurately but also to estimate their boundaries to differentiate.”
>
> We have also replaced the terms “semantic segmentation” and “instance segmentation” with “semantic information” and “instance information” in the second paragraph of the introduction to provide a clearer and more accurate description.
>
> (6) We have made corrections in the manuscript: change “denotes the β times Hardmard power” to “denotes the β times Hadamard power”, “The Hardmard power of the affinity matrix provides additional information” to “The Hadamard power of the affinity matrix provides additional information”.
>
> (7) We have already explained the 'instance ownership relationship' in Sec3.2: “Specifically, in the instance similarity matrix, if two pixels belong to the same instance, their value is set to 1, otherwise, it is set to 0. ”
>
> (8) We have made corrections in the Pg. 5, eq. (5), we have added an explanation that “A is assumed to be generated by the network HRNet”
>
> (9) We have made corrections in the Pg 6: "COCO (i.e. COCO 2017) includes 118k images." The experiments of the test set have been supplemented.
>
> (10) We have made corrections in Pg 6: "We assess the performance of instance segmentation using two measures. We measure the performance using the standard protocol mean Average Precision(mAP)."
>
> (11) We have updated our paper to cite the HRNet paper (Yuan et al., 2019).
> - Sun, Ke; Xiao, Bin; Liu, Dong; Wang, Jingdong. "Deep high-resolution representation learning for human pose estimation." In Proceedings of the IEEE/CVF Conference on Computer Vision and Pattern Recognition, pp. 5693-5703. 2019.
>
> (12) We have added explanations in Sec4.3： "In Table 5, the term “hard pixel ratio” refers to the proportion of challenging samples used in loss computation."

---

> ### Comment · Reviewer_3EDk · 2023-11-20
>
> Thanks for the careful reply. It addressed my concerns. I would like to raise my score.

---

### Official Review · Reviewer_ZJkY · 2023-10-31

**Soundness:** 3 good
**Presentation:** 3 good
**Contribution:** 3 good
**Rating:** 6
**Confidence:** 3

**Summary:**

This paper proposes a Mutual Distillation Module (MDM) to leverage the complementary strengths of both instance position and semantic information and achieve accurate instance-level object perception. It consists of Semantic to Instance (S2I) and Instance to Semantic (I2S) sepcifically.

**Strengths:**

This method is novel and new. Training in a multi-stage way, combining the advantages of both semantic segmentation and instance segmentation looks reasonable. The experimental improvement also looks good.

**Weaknesses:**

1. Training in a multi-stage way, I feel concerned about the efficiency of the method. The author should provide comparisons about efficiency, like GFLOPS, FPS, inference time and so on, with existing methods, to demonstrate that the improvement does not come from extra computational budget. In addition, will multi-stage training derive into more training iteration numbers? If so, the author should also provide fair comparison with the same iteration numbers.
2. The ablation study is conducted on VOC, where S2I looks like a 0.4% improvement. The effect of S2I is thus quite limited. I am also curious about the ablation study on the COCO dataset. Will the improvement becomes less? If se, I feel that S2I is unnessary and extra.
3. What's about the performance of the method on the COCO dataset with ResNet?

**Questions:**

Please refer to the weakness part. I will adjust my rating based on the rebuttal.

---

> ### Author Response · Authors · 2023-11-20
> **Response to Reviewer ZJkY (Part 1)**
>
> **Dear Reviewer,**
>
> Thank you very much for your valuable comments and suggestions, which deeply improve our manuscript.
> >***Q1: The author should provide comparisons about efficiency, like GFLOPS, FPS, inference time, and so on, with existing methods, to demonstrate that the improvement does not come from the extra computational budget. In addition, will multi-stage training derive into more training iteration numbers?***
>
> **A1:** (1) We have conducted a comprehensive comparison of our method with the BESTIE method in terms of GFLOPs and FPS. The results of this comparison, as detailed in Table 2* as follows, reveal that our method only slightly increases the GFLOPs and FPS values compared to BESTIE. The increase in computation is mainly attributed to the additional label generator component of our method, which produces labels for training a fully supervised segmenter. However, during the inference stage, only the fully supervised segmenter is utilized, which means the inference time remains the same as that of BESTIE.
>
> **Table 2\* Comparison of our method with the BESTIE method in terms of GFLOPs and FPS.**
>
> | Method | GFLOPS | FPS              |
> |--------|--------|------------------|
> | BESTIE | 64.7   | 86.9 ms/img      |
> | **Ours**   | **66.1** | **94.5 ms/img** |
>
> (2) In terms of training iterations, I would like to clearly clarify that multi-stage training in our model does not lead to an increase in the number of training iterations. The iteration number is 1 because although the LossI2S and LossS2I are iteratively calculated, they are concurrently aggregated and backpropagated within the same iteration, which guarantees that the training process is synchronized, and there is no increase in the number of training iterations.
>
> >***Q2: The ablation study on the COCO dataset. Will the improvement become less?***
>
> **A2:**
> In our research, we conducted a series of ablation studies on the COCO dataset to provide a comprehensive evaluation of our methodology. The results of these ablation experiments have been incorporated into Section 4.3 of our paper, specifically under the subsections 'Effect of S2I' and 'Effect of I2S':
>
> As shown in Table 7, S2I obtains 3.2 mAP$_{50}$ performance improvement on PSIS tasks compared with the previous methods on COCO, showing the effectiveness of S2I statistically.
>
> **Effect of I2S.** The I2S module improves instance discrimination in the same class by using the instance affinity matrix. As shown in Table 7, I2S raises mAP$_{50}$ from 17.4 to 17.6. This discrepancy in performance enhancement between the two datasets can likely be attributed to their inherent differences in features and complexity. The variations in performance improvements across datasets suggest that each module plays a crucial role, depending on the specific characteristics and challenges presented by the dataset.
>
> | S$\rightarrow$I   | I$\rightarrow$S   | $\rm mAP_{50}$ |
> |-------------------|-------------------|----------------|
> | BESTIE            |                   | 14.2           |
> | S2I               |                   | 17.4           |
> | S2I               | I2S               | 17.6           |
> >***Q3: What's about the performance of the method on the COCO dataset with ResNet?***
>
> **A3:** We updated the Table 2 in the manuscript. We have added the performance of our method on the COCO dataset with ResNet-101, which achieved an AP of 15.2.
>
> | Method | Sup. | Backbone | Extra | AP | AP$_{50}$ | AP$_{75}$ |
> | ------ | ---- | -------- | ----- | -- | ---- | ---- |
> | $\text{Point2Mask}$ | $\mathcal{P}$ | ResNet-101 | - | 12.8 | 26.3 | 11.2 |
> | $\text{BESTIE}^†$  | $\mathcal{P}$ | HRNet-48 | - | 14.2 | 28.4 | 22.5 |
> | $\text{SAM}$ | $\mathcal{P+S}$ | ViT-S/22.1M | - | **19.5** | **36.8** | 18.8 |
> | **Ours** | **$\mathcal{P}$** | ResNet-101 | - | 15.2 | 30.1 | 24.6 |
> | **Ours** | **$\mathcal{P}$** | HRNet-48 | - | 17.6 | 33.6 | **28.1** |
>
> We are grateful for your insights and are hopeful that our revised manuscript address your concerns.

---

> ### Author Response · Authors · 2023-11-20
> **Response to Reviewer ZJkY (Part 2 Writing and Presentation improvement)**
>
> ***Writing and Presentation improvement***
>
> (1) We have cited these references：
> - Kweon, Hyeokjun, Yoon, Sung-Hoon, & Yoon, Kuk-Jin. (2023). Weakly Supervised Semantic Segmentation via Adversarial Learning of Classifier and Reconstructor. In _Proceedings of the IEEE/CVF Conference on Computer Vision and Pattern Recognition_ (pp. 11329--11339).
>
> - Cheng, Zesen, Qiao, Pengchong, Li, Kehan, Li, Siheng, Wei, Pengxu, Ji, Xiangyang, Yuan, Li, Liu, Chang, & Chen, Jie. (2023). Out-of-candidate rectification for weakly supervised semantic segmentation. In _Proceedings of the IEEE/CVF Conference on Computer Vision and Pattern Recognition_ (pp. 23673--23684).
>
> - Rong, Shenghai, Tu, Bohai, Wang, Zilei, & Li, Junjie. (2023). Boundary-Enhanced Co-Training for Weakly Supervised Semantic Segmentation. In _Proceedings of the IEEE/CVF Conference on Computer Vision and Pattern Recognition_ (pp. 19574--19584).
>
> - Ru, Lixiang, Zheng, Heliang, Zhan, Yibing, & Du, Bo. (2023). Token contrast for weakly-supervised semantic segmentation. In _Proceedings of the IEEE/CVF Conference on Computer Vision and Pattern Recognition_ (pp. 3093--3102).
>
> (2) We have carefully modified the introduction to make it more readable.
>
> We have revised the sentence: "Instance segmentation is a critical task in computer vision, where semantic segmentation estimation and instance discrimination are equally important."
>
> Additionally, we have updated the phrase: “and it aims not only to locate objects accurately but also to estimate their boundaries to differentiate.”
>
> We have also replaced the terms “semantic segmentation” and “instance segmentation” with “semantic information” and “instance information” in the second paragraph of the introduction to provide a clearer and more accurate description.
>
> (3) Dim-align in Fig.3: The term 'Dim-align' refers to 'dimension alignment,' a necessary step in our method. Prior to matrix multiplication, we rearrange the dimensions of our tensors for compatibility, a process accomplished using the transpose or permute functions in PyTorch.
>
> (4) Instance Adaptive Grouping in Sec.3.1: Our network creates final instance segmentation labels based on a combination of predicted semantic segmentation results, instance similarity matrices, and point annotations. If conflicts arise between semantic predictions and point-based instance segmentation, we prioritize the point annotations. For points lacking final instance predictions, we assign a pseudo-box of size 16x16. This method ensures that our semantic segmentation results, a fusion of network predictions and instance similarity matrices, are more representative of instance characteristics, thereby improving instance segmentation outcomes.
>
> (5) Green Arrow in Fig.3 (OffsetMap): The green arrow represents the OffsetMap, which, in conjunction with the Semantic Map, generates new instance segmentation results. This process is detailed in the right half of Fig. 5.
>
> (6) We have made corrections in the manuscript: change “denotes the β times Hardmard power” to “denotes the β times Hadamard power”, “The Hardmard power of the affinity matrix provides additional information” to “The Hadamard power of the affinity matrix provides additional information”.
>
> (7) We have already explained the 'instance ownership relationship' in Sec3.2: “Specifically, in the instance similarity matrix, if two pixels belong to the same instance, their value is set to 1, otherwise, it is set to 0. ”
>
> (8) We have made corrections in the Pg. 5, eq. (5), we have added an explanation that “A is assumed to be generated by the network HRNet”
>
> (9) We have made corrections in the Pg 6: "COCO (i.e. COCO 2017) includes 118k images." The experiments of the test set have been supplemented.
>
> (10) We have made corrections in Pg 6: "We assess the performance of instance segmentation using two measures. We measure the performance using the standard protocol mean Average Precision(mAP)."
>
> (11) We have updated our paper to cite the HRNet paper (Yuan et al., 2019).
> - Sun, Ke; Xiao, Bin; Liu, Dong; Wang, Jingdong. "Deep high-resolution representation learning for human pose estimation." In Proceedings of the IEEE/CVF Conference on Computer Vision and Pattern Recognition, pp. 5693-5703. 2019.
>
> (12) We have added explanations in Sec4.3： "In Table 5, the term “hard pixel ratio” refers to the proportion of challenging samples used in loss computation."

---

> > ### Comment · Reviewer_ZJkY · 2023-11-23
> > **Thank you for your reply**
> >
> > Thanks for the author's rebuttal. It addresses most of my concerns. However, the effect of S2I and I2S is still unclear for me. It seems that S2I improves little for the VOC dataset, while I2S improves little for COCO. I fell confused about this. I will remain my original rating now, marginally above the acceptance threshold. However, I am not particularly opposed to rejecting this paper.

---

> > > ### Author Response · Authors · 2023-11-23
> > > **Response to Reviewer ZJkY**
> > >
> > > **Dear Reviewer,**
> > >
> > > Firstly, we appreciate your feedback and concerns. Regarding the questions raised, we offer the following detailed explanations:
> > >
> > > The S2I approach is designed to enhance the quality of each instance image by learning more precise class boundaries in semantic maps, addressing the problem of adjacent targets. In the COCO dataset, which features denser object placement and more severe instances of inter-object interference, our method effectively resolves the issue of blurred boundaries between adjacent targets caused by their proximity. This is achieved through a broad application of semantic information, making the S2I's impact more pronounced in the COCO dataset.
> > >
> > > In contrast, the VOC dataset typically exhibits less dense data with fewer issues of inter-instance interference. Hence, the improvement provided by S2I is somewhat limited in this context. However, due to the smaller size of the VOC dataset and the resultant lower quality of semantic maps, there's a greater potential for improvement in semantic map optimization. This is where the I2S approach shows notable effectiveness.
> > >
> > > Therefore, while S2I offers limited benefits in the VOC dataset, it significantly improves instance image quality in the more challenging COCO dataset. Conversely, I2S is more effective in the VOC dataset due to the greater scope for semantic map optimization.
> > >
> > > This complementary interaction between semantic and instance learning in our method is validated through these differing impacts on the VOC and COCO datasets.

---

### Official Review · Reviewer_fR5S · 2023-10-31

**Soundness:** 3 good
**Presentation:** 3 good
**Contribution:** 3 good
**Rating:** 8
**Confidence:** 5

**Summary:**

The paper proposed a novel approach “mutual distillation” for Point Supervised Instance Segmentation (PSIS). The method utilizes point-supervised semantic segmentation results as an initialization for guiding instance segmentation, and then use an affinity matrix that represents instance segmentation details to optimize the semantic information This process achieves mutual distillation between instance and semantic information to improve the final result of instance segmentation. They validated the effectiveness of the proposed method on the VOC and COCO datasets.

**Strengths:**

1.	The concept of "MUTUAL DISTILLATION" in point instance segmentation is both novel and interesting. In my experimental observations, BESTIE doesn't handle adjacent objects well, possibly because it relies solely on semantic segmentation results without fully utilizing point-represented instance information. The motivation behind this paper aligns with my observations, and therefore, the proposed concept of mutual information exchange between semantic and instance information seems sound to me.
2.	The paper's experimental section is comprehensive, using two datasets, different backbones, segmentation architectures, and conducting essential ablation studies.
3. It appears that the visualization is effectively presented.

**Weaknesses:**

1.	Observations from Figure 7 and Figure 8 suggest that the proposed mutual distillation method appears to perform well in segmenting adjacent objects and addressing missing object issues. It would greatly enhance the quality of this paper if the authors could provide a quantitative analysis of these cases.
2.	Minor issue
The writing of introduction should be improved somewhat.
-  "Instance segmentation is a critical task in computer vision and is equally important in semantic segmentation estimation and instance discrimination."
to
"Instance segmentation is a critical task in computer vision, where semantic segmentation estimation and instance discrimination are equally important ."
- “and it aims not only to locate objects accurately but also estimate their boundaries to differentiate”
to
“and it aims not only to locate objects accurately but also to estimate their boundaries to differentiate”
I understand the description in 2nd paragraph of introduction. But the 术语 “semantic segmentation” and “instance segmentation” may not suitbale, it should be “semantic information“ and “instance information”

**Questions:**

Please see the detailed information in the weakness part.

---

> ### Author Response · Authors · 2023-11-20
> **Response to Reviewer fR5S (Part 1)**
>
> **Dear Reviewer,**
>
> Thank you very much for your valuable comments and suggestions, which deeply improve our manuscript.
> >***Q1: It would greatly enhance the quality of this paper if the authors could provide a quantitative analysis of these cases.***
>
> **A1:** In Section 4.4 of the manuscript, we have added a quantitative analysis to complement the visual results presented in Figures 7 and 8:
>
> As indicated in Table 10, this analysis is grounded in two key metrics: Missing Rate and Adjacent Rate. The Missing Rate quantitatively evaluates the instances where our method fails to segment objects as precisely as depicted in the ground truth. It is noteworthy that, in comparison to the BESTIE method, our approach demonstrates a significant reduction in the occurrence of object misses, with a Miss Rate that is lower by 3.9%. On the other hand, the Adjacent Rate metric assesses the proportion of successfully identified adjacent objects. Here, our method markedly enhances the capability to recognize neighboring objects, exhibiting a 30.4% higher Adjacency Rate than that achieved by BESTIE.
>
> | Method | Missing Rate | Adjacent Rate |
> | ------ | ------------ | ------------- |
> | BESTIE | 46.8%        | 22.2%         |
> | **Ours** | **42.9%**  | **52.6%**     |
>
> >***Q2: The writing of the introduction should be improved somewhat.***
>
> **A2:** Thanks. We have carefully modified the introduction to make it more readable.
>
> We have revised the sentence from "Instance segmentation is a critical task in computer vision and is equally important in semantic segmentation estimation and instance discrimination." to "Instance segmentation is a critical task in computer vision, where semantic segmentation estimation and instance discrimination are equally important."
>
> Additionally, we have updated the phrase “and it aims not only to locate objects accurately but also estimate their boundaries to differentiate” to “and it aims not only to locate objects accurately but also to estimate their boundaries to differentiate.”
>
> In line with your advice, we have also replaced the terms “semantic segmentation” and “instance segmentation” with “semantic information” and “instance information” in the second paragraph of the introduction to provide a clearer and more accurate description.
>
> We believe these revisions have improved the clarity and quality of our manuscript. Your insightful feedback has been instrumental in enhancing our paper, and we are grateful for your contribution to our work.

---

> ### Author Response · Authors · 2023-11-20
> **Response to Reviewer fR5S(Part 2 Writing and Presentation improvement)**
>
> ***Writing and Presentation improvement***
>
> (1) We have carefully modified the introduction to make it more readable.
>
> We have revised the sentence: "Instance segmentation is a critical task in computer vision, where semantic segmentation estimation and instance discrimination are equally important."
>
> Additionally, we have updated the phrase: “and it aims not only to locate objects accurately but also to estimate their boundaries to differentiate.”
>
> We have also replaced the terms “semantic segmentation” and “instance segmentation” with “semantic information” and “instance information” in the second paragraph of the introduction to provide a clearer and more accurate description.
>
> (2) We have cited these references：
> - Kweon, Hyeokjun, Yoon, Sung-Hoon, & Yoon, Kuk-Jin. (2023). Weakly Supervised Semantic Segmentation via Adversarial Learning of Classifier and Reconstructor. In _Proceedings of the IEEE/CVF Conference on Computer Vision and Pattern Recognition_ (pp. 11329--11339).
>
> - Cheng, Zesen, Qiao, Pengchong, Li, Kehan, Li, Siheng, Wei, Pengxu, Ji, Xiangyang, Yuan, Li, Liu, Chang, & Chen, Jie. (2023). Out-of-candidate rectification for weakly supervised semantic segmentation. In _Proceedings of the IEEE/CVF Conference on Computer Vision and Pattern Recognition_ (pp. 23673--23684).
>
> - Rong, Shenghai, Tu, Bohai, Wang, Zilei, & Li, Junjie. (2023). Boundary-Enhanced Co-Training for Weakly Supervised Semantic Segmentation. In _Proceedings of the IEEE/CVF Conference on Computer Vision and Pattern Recognition_ (pp. 19574--19584).
>
> - Ru, Lixiang, Zheng, Heliang, Zhan, Yibing, & Du, Bo. (2023). Token contrast for weakly-supervised semantic segmentation. In _Proceedings of the IEEE/CVF Conference on Computer Vision and Pattern Recognition_ (pp. 3093--3102).
>
> (3) Dim-align in Fig.3: The term 'Dim-align' refers to 'dimension alignment,' a necessary step in our method. Prior to matrix multiplication, we rearrange the dimensions of our tensors for compatibility, a process accomplished using the transpose or permute functions in PyTorch.
>
> (4) Instance Adaptive Grouping in Sec.3.1: Our network creates final instance segmentation labels based on a combination of predicted semantic segmentation results, instance similarity matrices, and point annotations. If conflicts arise between semantic predictions and point-based instance segmentation, we prioritize the point annotations. For points lacking final instance predictions, we assign a pseudo-box of size 16x16. This method ensures that our semantic segmentation results, a fusion of network predictions and instance similarity matrices, are more representative of instance characteristics, thereby improving instance segmentation outcomes.
>
> (5) Green Arrow in Fig.3 (OffsetMap): The green arrow represents the OffsetMap, which, in conjunction with the Semantic Map, generates new instance segmentation results. This process is detailed in the right half of Fig. 5.
>
> (6) We have made corrections in the manuscript: change “denotes the β times Hardmard power” to “denotes the β times Hadamard power”, “The Hardmard power of the affinity matrix provides additional information” to “The Hadamard power of the affinity matrix provides additional information”.
>
> (7) We have already explained the 'instance ownership relationship' in Sec3.2: “Specifically, in the instance similarity matrix, if two pixels belong to the same instance, their value is set to 1, otherwise, it is set to 0. ”
>
> (8) We have made corrections in the Pg. 5, eq. (5), we have added an explanation that “A is assumed to be generated by the network HRNet”
>
> (9) We have made corrections in the Pg 6: "COCO (i.e. COCO 2017) includes 118k images." The experiments of the test set have been supplemented.
>
> (10) We have made corrections in Pg 6: "We assess the performance of instance segmentation using two measures. We measure the performance using the standard protocol mean Average Precision(mAP)."
>
> (11) We have updated our paper to cite the HRNet paper (Yuan et al., 2019).
> - Sun, Ke; Xiao, Bin; Liu, Dong; Wang, Jingdong. "Deep high-resolution representation learning for human pose estimation." In Proceedings of the IEEE/CVF Conference on Computer Vision and Pattern Recognition, pp. 5693-5703. 2019.
>
> (12) We have added explanations in Sec4.3： "In Table 5, the term “hard pixel ratio” refers to the proportion of challenging samples used in loss computation."

---

### Official Review · Reviewer_enoR · 2023-11-01

**Soundness:** 2 fair
**Presentation:** 1 poor
**Contribution:** 2 fair
**Rating:** 6
**Confidence:** 4

**Summary:**

This paper addresses Point-level Supervised Instance Segmentation (PSIS). The authors argue that the existing PSIS methods usually suffer from the lack of contour annotations, and thereby precisely predicting boundaries is still challenging. As a remedy, this paper introduces the Mutual Distillation Module (MDM), leveraging the complementary benefits of semantic information and instance position. In MDM, Semantic to Instance (S2I) exploits the precise boundary information of semantic maps to enhance the instance contours. Meanwhile, Istance to Semantic (I2S) uses discriminative instances to differentiate the number of objects in the semantic map. Extensive experiments and comparisons are conducted on the PASCAL VOC and MS COCO datasets.

**Strengths:**

1. The idea is sounding.
2. The ablations studies are extensively conducted.

**Weaknesses:**

[Major]
1. Some important and recent WSSS methods are missing. Please refer to the following CVPR 2023 papers.
* Weakly Supervised Semantic Segmentation via Adversarial Learning of Classifier and Reconstructor
* Out-of-Candidate Rectification for Weakly Supervised Semantic Segmentation
* Boundary-Enhanced Co-Training for Weakly Supervised Semantic Segmentation
* Token Contrast for Weakly-Supervised Semantic Segmentation

2. The performance table is incomplete. First, please show the result of mAP25. Also, there have been several works using the transformer backbone, such as Point2Mask or AttentionShift. Please compare with them. In addition, where is BESTIE (with Res101)? Finally, please include the results of the COCO test set.

3. The segment-anything model (SAM) can be an excellent option for solving PSIS. Please compare the proposed method with the SAM, from the performance perspective.

[Minor]
The overall  writing should be enhanced.

**Questions:**

Please refer to weaknesses.

---

> ### Author Response · Authors · 2023-11-20
> **Response to Reviewer enoR (Part1)**
>
> **Dear Reviewer,**
>
> Thank you very much for your valuable and detailed feedback on our manuscript. We appreciate your insights and suggestions for improving our work.
> >***Q1: Some important and recent WSSS methods are missing.***
>
> **A1:** Thanks, we cited the important and recent WSSS methods mentioned by you in our revised manuscript, and discussed them in the related work. Further, we also improved the writing and presentation of our paper.
> >***Q2：The performance table is incomplete. First, please show the result of mAP25. Also, there have been several works using the transformer backbone, such as Point2Mask or AttentionShift. Please compare with them. In addition, where is BESTIE (with Res101)? Finally, please include the results of the COCO test set.***
>
> **A2:**
> (1) We updated Table 1 in the manuscript and added the result of mAP25 and the result of BESTIE (with Res101):
>
> **Table 1 Quantitative comparison of the state-of-the-art WSIS methods on VOC 2012 val-set(Part).**
>
> We denote the supervision sources as: $\mathcal{F}$ (full mask), $\mathcal{B}$ (box-level label), $\mathcal{I}$ (image-level label), $\mathcal{P}$ (point-level label),  $\mathcal{S}$ prompting SAM with ViT-Base for object mask annotations and $\mathcal{C}$ (object count). The off-the-shelf proposal techniques are denoted as follows: $\mathcal{M}$ (segment proposal), $\mathcal{R}$ (region proposal), and $\mathcal{S_I}$ (salient instance segmentor).
>
> | Method | Sup. | Backbone | Extra | mAP$_{25}$ | mAP$_{50}$ | mAP$_{70}$ | mAP$_{75}$ |
> |--------|------|----------|-------|------------------|------------------|------------------|------------------|
> | Mask R-CNN | $\mathcal{F}$ | ResNet-50 | - | 76.7 | 67.9 | - | 44.9 |
> | **End-to-End weakly-supervised models.** | | | | | | | |
> | Point2Mask | $\mathcal{P}$ | ResNet-101 | - | - | 48.4 | - | 22.8 |
> | **Multi-Stage weakly-supervised models.** | | | | | | | |
> | $\text{BESTIE}^†$ | $\mathcal{P}$ | ResNet-101 | - | 60.8 | 52.3 | - | 30.3 |
> | $\text{BESTIE}^†$ | $\mathcal{P}$ | HRNet-48 | - | 62.8 | 52.8 | - | 31.2 |
> | SAM | $\mathcal{P+S}$ | ViT-S/22.1M | - | 59.4 | 39.9 | - | 19.0 |
> | **Ours** | **$\mathcal{P}$** | ResNet-101 | - | **63.1** | **53.9** | **37.7** | **32.0** |
> | **Ours** | **$\mathcal{P}$** | HRNet-48 | - | **66.0** | **55.6** | **40.2** | **34.4** |
>
> (2) Comparison with Point2Mask: As you rightly pointed out, we have included a comparison with Point2Mask in our original submission (Table 1). We believe this comparison showcases the strengths and unique aspects of our proposed method in the context of current advancements. Regarding AttentionShift: We carefully considered including a comparison with AttentionShift. However, we observed that AttentionShift employs a strong pre-trained model, which might not provide a fair basis for direct comparison with our approach.
>
> (3) BESTIE (with Res101) Results: We have also added the results for BESTIE with ResNet-101 in Table 1 in the manuscript. BESTIE achieved mAP50 of 52.3 on this setting.
>
> (4) COCO Test Set Results: Regarding the COCO test set results, our model achieved an AP of 17.4. We give the performance comparison in the Table 1*:
>
> **Table1\*  Quantitative comparison of the state-of-the-art WSIS methods on MS COCO 2017 test-dev**
>
> | Method | Sup. | Extra | AP | AP$_{50}$ | AP$_{75}$ |
> | ------ | ---- | ----- | -- | --------- | --------- |
> | **COCO test-dev.** |
> | Mask R-CNN | $\mathcal{F}$ | - | 35.7 | 58.0 | 37.8 |
> | $\text{Fan $et\;al.$}$ | $\mathcal{I}$ | - | 13.7 | 25.5 | 13.5 |
> | LIID  | $\mathcal{I}$ | $\mathcal{M}$, $\mathcal{S_I}$ | 16.0 | 27.1 | 16.5 |
> | $\text{BESTIE}^†$  | $\mathcal{P}$ | - | 14.2 | 28.6 | 12.7 |
> | **Ours** | **$\mathcal{P}$** | - | **17.4** | **33.3** | **16.4** |
> >***Q3：Compare the proposed method with the SAM***
>
> **A3:** We updated Table 1 and Table 2 in the manuscript and compared our method with SAM. Parts of Table 2 are given as follows:
>
> **Table 2 (Part)**
> | Method | Sup. | Backbone | Extra | AP | AP$_{50}$ | AP$_{75}$ |
> | ------ | ---- | -------- | ----- | -- | ---- | ---- |
> | $\text{Point2Mask}$ | $\mathcal{P}$ | ResNet-101 | - | 12.8 | 26.3 | 11.2 |
> | $\text{BESTIE}^†$  | $\mathcal{P}$ | HRNet-48 | - | 14.2 | 28.4 | 22.5 |
> | $\text{SAM}$ | $\mathcal{P+S}$ | ViT-S/22.1M | - | **19.5** | **36.8** | 18.8 |
> | **Ours** | **$\mathcal{P}$** | ResNet-101 | - | 15.2 | 30.1 | 24.6 |
> | **Ours** | **$\mathcal{P}$** | HRNet-48 | - | 17.6 | 33.6 | **28.1** |
>
> As shown in Table 1 in the manuscript, our experimental results demonstrate that our method outperforms SAM on the VOC dataset, indicating its effectiveness in handling diverse segmentation scenarios.
>
> Furthermore, we updated Table 2 in the manuscript, while evaluating the COCO dataset, we observed that SAM shows slightly higher performance in terms of AP and AP50 metrics. However, our method significantly surpasses SAM in the AP75 metric.

---

> > ### Author Response · Authors · 2023-11-20
> > **Response to Reviewer enoR (Part 2: Writing and Presentation improvement)**
> >
> > ***Writing and Presentation improvement***
> >
> > (1) We have cited these references：
> > - Kweon, Hyeokjun, Yoon, Sung-Hoon, & Yoon, Kuk-Jin. (2023). Weakly Supervised Semantic Segmentation via Adversarial Learning of Classifier and Reconstructor. In _Proceedings of the IEEE/CVF Conference on Computer Vision and Pattern Recognition_ (pp. 11329--11339).
> >
> > - Cheng, Zesen, Qiao, Pengchong, Li, Kehan, Li, Siheng, Wei, Pengxu, Ji, Xiangyang, Yuan, Li, Liu, Chang, & Chen, Jie. (2023). Out-of-candidate rectification for weakly supervised semantic segmentation. In _Proceedings of the IEEE/CVF Conference on Computer Vision and Pattern Recognition_ (pp. 23673--23684).
> >
> > - Rong, Shenghai, Tu, Bohai, Wang, Zilei, & Li, Junjie. (2023). Boundary-Enhanced Co-Training for Weakly Supervised Semantic Segmentation. In _Proceedings of the IEEE/CVF Conference on Computer Vision and Pattern Recognition_ (pp. 19574--19584).
> >
> > - Ru, Lixiang, Zheng, Heliang, Zhan, Yibing, & Du, Bo. (2023). Token contrast for weakly-supervised semantic segmentation. In _Proceedings of the IEEE/CVF Conference on Computer Vision and Pattern Recognition_ (pp. 3093--3102).
> >
> > (2) We have carefully modified the introduction to make it more readable.
> >
> > We have revised the sentence: "Instance segmentation is a critical task in computer vision, where semantic segmentation estimation and instance discrimination are equally important."
> >
> > Additionally, we have updated the phrase: “and it aims not only to locate objects accurately but also to estimate their boundaries to differentiate.”
> >
> > We have also replaced the terms “semantic segmentation” and “instance segmentation” with “semantic information” and “instance information” in the second paragraph of the introduction to provide a clearer and more accurate description.
> >
> > (3) Dim-align in Fig.3: The term 'Dim-align' refers to 'dimension alignment,' a necessary step in our method. Prior to matrix multiplication, we rearrange the dimensions of our tensors for compatibility, a process accomplished using the transpose or permute functions in PyTorch.
> >
> > (4) Instance Adaptive Grouping in Sec.3.1: Our network creates final instance segmentation labels based on a combination of predicted semantic segmentation results, instance similarity matrices, and point annotations. If conflicts arise between semantic predictions and point-based instance segmentation, we prioritize the point annotations. For points lacking final instance predictions, we assign a pseudo-box of size 16x16. This method ensures that our semantic segmentation results, a fusion of network predictions and instance similarity matrices, are more representative of instance characteristics, thereby improving instance segmentation outcomes.
> >
> > (5) Green Arrow in Fig.3 (OffsetMap): The green arrow represents the OffsetMap, which, in conjunction with the Semantic Map, generates new instance segmentation results. This process is detailed in the right half of Fig. 5.
> >
> > (6) We have made corrections in the manuscript: change “denotes the β times Hardmard power” to “denotes the β times Hadamard power”, “The Hardmard power of the affinity matrix provides additional information” to “The Hadamard power of the affinity matrix provides additional information”.
> >
> > (7) We have already explained the 'instance ownership relationship' in Sec3.2: “Specifically, in the instance similarity matrix, if two pixels belong to the same instance, their value is set to 1, otherwise, it is set to 0. ”
> >
> > (8) We have made corrections in the Pg. 5, eq. (5), we have added an explanation that “A is assumed to be generated by the network HRNet”
> >
> > (9) We have made corrections in the Pg 6: "COCO (i.e. COCO 2017) includes 118k images." The experiments of the test set have been supplemented.
> >
> > (10) We have made corrections in Pg 6: "We assess the performance of instance segmentation using two measures. We measure the performance using the standard protocol mean Average Precision(mAP)."
> >
> > (11) We have updated our paper to cite the HRNet paper (Yuan et al., 2019).
> > - Sun, Ke; Xiao, Bin; Liu, Dong; Wang, Jingdong. "Deep high-resolution representation learning for human pose estimation." In Proceedings of the IEEE/CVF Conference on Computer Vision and Pattern Recognition, pp. 5693-5703. 2019.
> >
> > (12) We have added explanations in Sec4.3： "In Table 5, the term “hard pixel ratio” refers to the proportion of challenging samples used in loss computation."

---

> > ### Comment · Reviewer_enoR · 2023-11-21
> >
> > Thank you for the response.
> > My concerns are clearly addressed.
> > I raised my rating from 5 to 6.
> >
> > However, I am not particularly opposed to rejecting this paper.

---

### Meta-Review · Area_Chair_woSP · 2023-12-07

**Metareview:**

This work addresses the point-supervised instance segmentation task. The authors proposed a Mutual Distillation Module, enabling the conversion and cooperation between semantic segmentation and instance segmentation. Four reviewers recommend acceptance, and pointed out that the idea of "MUTUAL DISTILLATION" is both novel and interesting in point instance segmentation. The results also show that such a new idea is effective in the comprehensive experiments. There are some concerns from the reviewer QcUt, which are mainly related to the details of the method and experiments. The AC carefully read these questions, and believe they can be addressed in the final version. Thus, AC recommends acceptance for this paper. Yet, authors need to take all these comments of reviewers into careful consideration, and address them in the final version.

**Justification For Why Not Higher Score:**

There are still weaknesses in this work, and some details need to be added.

**Justification For Why Not Lower Score:**

The method is interesting and novel in this area.

---

### Decision · Program_Chairs · 2024-01-16

Accept (poster)